# CryoEM structure of the type IVa pilus secretin required for natural competence in *Vibrio cholerae*

Sara J. Weaver [ID] [1,5], Davi R. Ortega [ID] [2], Matthew H. Sazinsky [ID] [3], Triana N. Dalia [ID] [4], Ankur B. Dalia [ID] [4] & Grant J. Jensen [ID] [5✉]

Natural transformation is the process by which bacteria take up genetic material from their environment and integrate it into their genome by homologous recombination. It represents one mode of horizontal gene transfer and contributes to the spread of traits like antibiotic resistance. In *Vibrio cholerae*, a type IVa pilus (T4aP) is thought to facilitate natural transformation by extending from the cell surface, binding to exogenous DNA, and retracting to thread this DNA through the outer membrane secretin, PilQ. Here, we use a functional tagged allele of VcPilQ purified from native *V. cholerae* cells to determine the cryoEM structure of the VcPilQ secretin in amphipol to ~2.7 Å. We use bioinformatics to examine the domain architecture and gene neighborhood of T4aP secretins in Proteobacteria in comparison with VcPilQ. This structure highlights differences in the architecture of the T4aP secretin from the type II and type III secretion system secretins. Based on our cryoEM structure, we design a series of mutants to reversibly regulate VcPilQ gate dynamics. These experiments support the idea of VcPilQ as a potential druggable target and provide insight into the channel that DNA likely traverses to promote the spread of antibiotic resistance via horizontal gene transfer by natural transformation.

[1] Division of Chemistry and Chemical Engineering, California Institute of Technology, 1200 E. California Blvd, Pasadena, CA 91125, USA. [2] Division of Biology and Biological Engineering and Howard Hughes Medical Institute, California Institute of Technology, 1200 E. California Blvd, Pasadena, CA 91125, USA. [3] Department of Chemistry, Pomona College, 333N. College Way, Claremont, CA 91711, USA. [4] Department of Biology, Indiana University, 107S. Indiana Avenue, Bloomington, IN 47405, USA. [5] Present address: Howard Hughes Medical Institute, David Geffen School of Medicine, Departments of Biological Chemistry and Physiology, University of California Los Angeles, 615 Charles E Young Drive South, Los Angeles, CA 90095, USA. ✉email: jensen@caltech.edu

Horizontal gene transfer, or the ability of microorganisms to directly share DNA with one another, facilitates rapid evolution, can contribute to the development of antibiotic resistance, promotes the spread of virulence factors, and allows bacterial pathogens to rapidly evade host immune response[1]. A clear understanding of the mechanisms of horizontal gene transfer can aid the development of tools in the fight against antibiotic resistance.

One mechanism of horizontal gene transfer is natural transformation, where a competent bacterium can take up DNA from its environment and then maintain this exogenous genetic material, either as a plasmid or by integrating it into its genome by homologous recombination[2]. Many bacteria utilize a type IVa pilus (T4aP) nanomachine to take up genetic material[3,4]. Inter-genus transformation can result in the development of mosaic alleles that confer antibiotic resistance, and has been demonstrated in a variety of genera, including *Streptococcus*, *Neisseria*, and *Actinobacter*[5,6]. Additionally, natural transformation of large regions of DNA can induce serotype switching in *Vibrio cholerae*[7,8] and in *Streptococcus pneumoniae*[9]. Thus, preventing horizontal gene transfer represents a unique approach to mitigate the spread of antibiotic resistance and virulence in bacterial pathogens.

Here, we focus our attention to the structural biology of natural transformation in the gram-negative bacterium *V. cholerae*. *V. cholerae* is the causative agent of the diarrheal disease cholera[10]. Since 1817, cholera has spread globally in seven pandemics that each feature strains of distinct characteristics. Humans typically encounter members of the *Vibrionaceae* family through contaminated water or contaminated shellfish[11,12]. *V. parahaemolyticus* and *V. vulnificus* are common causes of shellfish-borne illness[13,14]. Type IV pilus (T4P) protein filaments are important in *Vibrionaceae* family pathogenicity because they mediate natural transformation, adhesion, biofilm formation, and colonization of their hosts[15–20]. The genomes of *V. cholerae*, *V. parahaemolyticus*, and *V. vulnificus* all contain two T4aP systems: mannose-sensitive hemagglutinin (MSHA) pili and chitin-regulated T4aP used for natural competence[21]; the latter of which allows for chitin-induced natural transformation[22–26]. Of these species, *V. cholerae* is the most genetically tractable and has emerged as a model system for studying natural transformation and bacterial pili.

Natural transformation in *V. cholerae* is tightly regulated, and is induced when these bacteria are grown on chitin, a biopolymer found in the exoskeletons of crustaceans, in their aquatic environment[23]. Chitin indirectly promotes expression of the master regulator of competence TfoX[27,28], which, in turn, induces expression of the T4aP needed for DNA uptake[23,29,30]. The T4aP machinery requires four elements: an inner membrane pilus assembly complex, cytoplasmic motors to extend and retract the pilus, an outer membrane pore (the secretin), and the pilin sub-units that compose the pilus itself[4]. The T4aP facilitates environmental DNA uptake by extending and retracting from the cell surface through a large, outer membrane secretin pore called PilQ[30,31]. To mediate DNA uptake, the pilus likely translocates DNA across the membrane through the PilQ secretin[4]. Thus, the PilQ secretin represents a potential target to thwart this mechanism of horizontal gene transfer.

The *V. cholerae* T4aP secretin PilQ is a member of the bacterial secretin superfamily[32,33]. Secretins are found in the type II secretion system (T2SS), the T3SS, the T4P machine, and filamentous phage[34]. Bacterial secretins are united by a common C-terminal secretin domain, which oligomerizes to form a large pore in the outer membrane[35]. While the C-terminal secretin domain is remarkably similar across these secretion systems, the N-terminal region varies, which may be related to the specialization of different secretins[36]. The T2SS exports periplasmic folded proteins to the extracellular space. The T3SS uses a "needle and syringe" to export cytosolic effector proteins outside of the cell, or directly into another cell. The T4P have a broad range of functionality, including DNA uptake, twitching motility, biofilm formation, and adhesion[15,16,30,37]. During natural transformation, the T4aP extends and retracts a filament to take up DNA cargo.

The structure of various T4P secretins has been examined[38–54], including structures of PilQ from *Neisseria meningitidis*[45], *Pseudomonas aeruginosa*[46], and *Thermus thermophilus*[47], although no work has provided high-resolution details sufficient to model the passage of DNA or to design inhibitors of this potential drug target. Thus, structural information about the T4P secretins has mainly been inferred from the related but distinct T2SS and T3SS secretins[55–61]. While this paper was under review, a 4.3 Å structure of the T4P PilQ from *P. aeruginosa* was reported[62].

Here, we present the structure of the *V. cholerae* T4aP secretin PilQ to ~2.7 Å using a fully functional, His-tagged allele that we express and purify from the native bacterium. We perform sequence analysis on the Proteobacteria T4aP secretins to put this structure into context. Our work highlights differences between T4aP, T2SS, and T3SS secretins, and emphasizes the need for structures of different secretin family members. In particular, we discuss differences and remaining puzzles, including how the pilus could be accommodated within VcPilQ during natural transformation and what part/s of the secretin if any penetrate the outer membrane. Finally, we report structure-inspired designs of cysteine pair mutants that allowed us to reversibly inhibit pilus assembly and natural transformation, presumably by sealing the secretin gate. These experiments support the designation of VcPilQ as a druggable target, and more broadly demonstrate how cysteine pair mutations can be employed to study the activity of bacterial secretins.

## Results

**Single-particle cryoEM of the T4aP secretin PilQ**. To ensure properly folded and fully functional T4aP machinery, we chose to purify PilQ from *V. cholerae* rather than a recombinant system. A chromosomal mutation was made to add a deca-histidine tag to the N-terminus of PilQ. Here, expression of the T4aP system was induced via ectopic expression of TfoX using an IPTG-regulated promoter ($P_{tac}$-*tfoX*)[63,64]. The bacteria expressing His-tagged VcPilQ retained wild-type levels of natural transformation (Supplementary Fig. 1a). Previous work demonstrated that similar N-terminal tags allowed for functional T4aP activity[31].

Secretins are detergent- and heat-resistant multimers[65,66]. Coomassie staining and western blotting of VcPilQ in amphipol run under denaturing conditions on a SDS Page gel showed monomer, the multimer, and some low molecular weight species in the purified sample (Supplementary Fig. 1b). The size difference between these contaminants (<100 kDa) and the VcPilQ multimer (~860 kDa) made it easy to distinguish VcPilQ from the milieu in electron micrographs (Supplementary Fig. 2a).

Here we report the high-resolution structure of the purified T4aP secretin VcPilQ by single-particle cryoEM (Fig. 1a and Table 1). The cryoEM data processing steps are summarized as a flow chart in Supplementary Fig. 3. The cryoEM structure (C14 symmetry, overall resolution of 2.7 Å at Fourier shell correlation (FSC) of 0.143, and 3 Å at FSC of 0.5) reached sufficient, isotropic resolution to recognize and model residues 160–571 of VcPilQ, which includes the N0, N3, and secretin domains (Fig. 1a, Supplementary Fig. 4, and Supplementary Movie 1)[67,68]. Mass spectrometry analysis demonstrated 65% sequence coverage, with fragments identified in each domain of

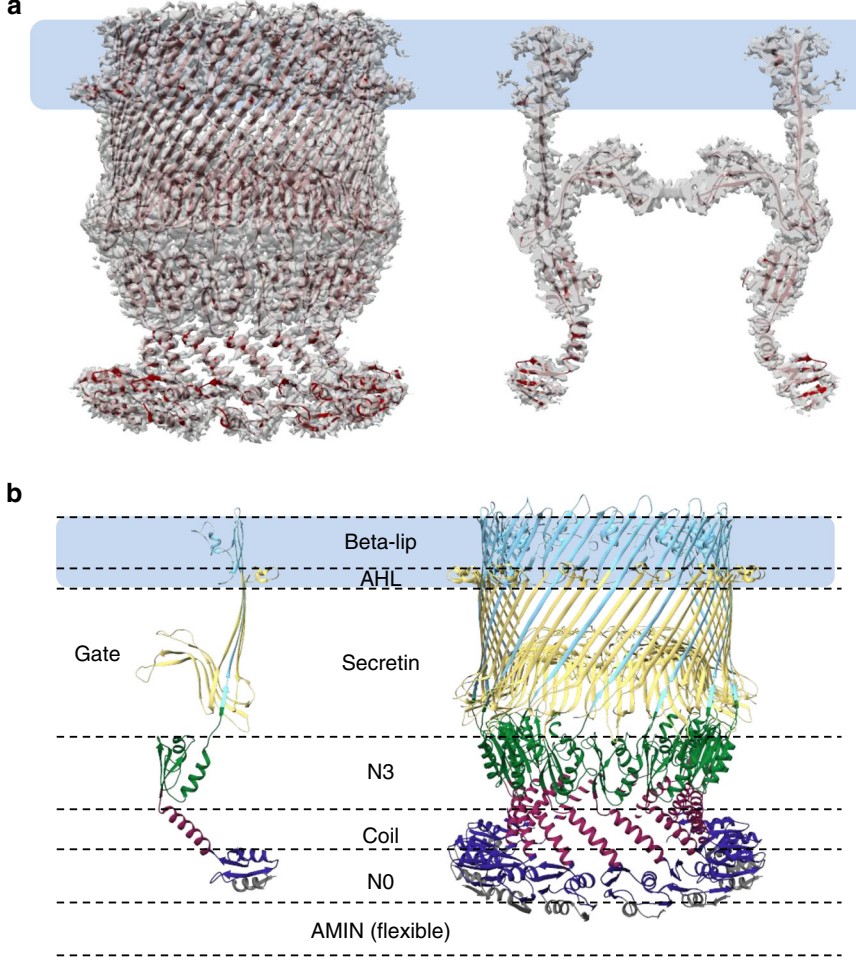

**Fig. 1 CryoEM structure of the *V. cholerae* type IV competence pilus secretin PilQ. a** Model (cartoon in dark red) and cryoEM density (dark gray transparent) of symmetrized *Vibrio cholerae* PilQ (VcPilQ) is shown from the side (left) and cut through the center (right). The putative outer membrane region is represented in a light blue rectangle. **b** The atomic model of the VcPilQ multimer is shown (right). One chain is shown by itself (left). The putative outer membrane region is represented in a light blue rectangle. Dashed lines represent the different domains of the VcPilQ structure: AMIN (not shown), N0 (gray/dark blue), coil (purple), N3 (green), secretin (yellow), beta lip (sky blue). The gate region is labeled on the monomer.

the folded protein from residues 50–567 (Supplementary Fig. 1c). Residues 50–159 were represented in the mass spectrometry analysis (Supplementary Fig. 1c), but were not clearly resolved in the cryoEM structure. A hazy density is seen in some 2D classes (Supplementary Fig. 2c, asterisk) in the region residues 50–159 would likely be present. These residues contain an AMIN Pfam domain, which is made up of two antiparallel beta sheets and is thought to bind peptidoglycan[69,70].

In each VcPilQ monomer, four beta strands come together to form a beta sheet (Fig. 1a). Once assembled, VcPilQ forms a 56-strand beta barrel. Inside the barrel, two further beta hairpins (beta strand-turn-beta strand each) form a gate. These regions match the topology of other secretin structures (Fig. 2a–c). In each case, the root-mean-square deviation (RMSD) calculated over the protein domains present in both structures (N3, secretin, and beta lip domains) was about 1 Å (Supplementary Fig. 5a–e). The outer membrane regions differ significantly, however, in both angle and membrane spanning distance, as discussed in more detail below.

Only one of the seven T2SS secretins and one of the T3SS secretins published structures resolve the N0 domain[59,61]. Here we can resolve the N0 domain (residues 160–227) of VcPilQ to 4–7 Å local resolution (Supplementary Fig. 4e–h), which allowed us to build a model based on homology to previously solved

structures of N0 domains (the cross-linked *K. oxytoca* PulD[59] and *Salmonella typhimurium* InvG[61]), plus a crystal structure of an isolated N0 domain from *N. meningitidis* PilQ (4AR0)[45].

In VcPilQ, a 32 Å alpha helix follows the N0 domain (Fig. 1b). None of the T2SS or T3SS structures contain a helical coil to link N-terminal domains; instead their periplasmic protein domains are linked by unstructured loops (Fig. 2). Following the end of this helix, the protein chain changes direction (~104° angle) as the coil flows into the N3 domain (Fig. 1b). This dramatically reduces the channel diameter, from 90 Å at the bottom of the N0 domain to 60 Å across the N3 domain (Supplementary Fig. 7a). In the T2SS and T3SS structures, the diameter of the channel is relatively constant.

While their structures are similar, the electrostatic characteristics of the inner surfaces of the T4P, T2SS, and T3SS secretins vary (Fig. 2d). The inner beta lip, N3, and coil regions of VcPilQ (Fig. 1b) are all negatively charged (Fig. 2d). The gate region of VcPilQ is also negatively charged (Supplementary Fig. 5f, g). In contrast, in the *V. cholerae* GspD and *E. coli* K12 GspD structures, there are alternating negatively and weakly positively charged regions (Fig. 2d).

**The putative outer membrane region of VcPilQ is thicker than T2SS secretins.** The secretin amphipathic helix lip (AHL) is

**Table 1 Summary of single-particle data collection, 3D reconstruction, and model refinement.**

| | |
|---|---|
| **Imaging parameters and 3D reconstruction** | |
| Acceleration voltage (kV) | 300 |
| Magnification (×) | 81,000 |
| Pixel size (Å) | 1.104 |
| Frame rate (s⁻¹) | 0.092 |
| Exposure time (s) | 3.7 |
| Total exposure (e⁻/Å) | 60 |
| Particles | |
| Micrographs used for selection | 2,510 |
| Defocus range (µm) | −0.5 to −3.5 |
| Windowed | 252,319 |
| In final 3D reconstruction | 100,543 |
| Resolution | |
| "Gold-standard" at FSC 0.5 (Å) | 3.0 Å |
| "Gold-standard" at FSC 0.143 (Å) | 2.7 Å |
| Map-sharpening B factor (Å²) | −69 |
| **Model refinement** | |
| Resolution in phenix.real_space_refine (Å) | 3.0 |
| Model-to-map fit (CC_mask) | 0.745 |
| Number of atoms/residues/molecules | |
| NCS restrained chains | 14 |
| Protein atoms, residues (per chain) | 43,736, 412 |
| Ramachandran angles (%) | |
| Favored | 92.21 |
| Allowed | 7.79 |
| Outliers | 0 |
| r.m.s. deviations | |
| Bond lengths (Å) | 0.006 |
| Bond angles (°) | 0.847 |
| Molprobity | |
| Score | 2.81 |
| Clashscore | 10.53 |
| Rotamer outliers (%) | 10.91 |
| EMRinger score | 3.13 |

thought to be a key determinant for membrane insertion and acts as the lower boundary of the outer membrane region, but the exact location of the upper boundary is less clear[34,56,60,61]. Recently, Ghosal et al.[71] solved the in situ structure of the *Legionella pneumophila* T2SS by sub-tomogram averaging and demonstrated that the putative outer membrane thickness reported in the single-particle cryoEM structures of detergent-solubilized T2SS secretins (~2.5 nm) is significantly smaller than real membranes (5–7 nm)[71]. In our VcPilQ atomic model, the distances between the bottom of the AHL to the top of the beta strands is about 3 nm, which is substantially taller than the same region in the previously published T2SS structures (Fig. 2c).

To investigate more than just the residue locations, we generated an inverted mask based on the atomic model density of VcPilQ and multiplied by the empirical, unsharpened cryoEM density to remove density accounted for by the atomic model ("Methods" and Supplementary Fig. 2e). The product reflects unmodeled density in the cryoEM map that is not accounted for by the atomic model (Supplementary Fig. 2e). VcPilQ was solubilized in n-Dodecyl-b-d-Maltoside (DDM) and then exchanged into amphipol, so the unaccounted for density in our structure could be some combination of DDM, amphipol, protein, lipid, and noise. Rather than select one cryoEM threshold to visualize the membrane thickness at, we chose to measure the membrane thickness as a series of thresholds so that a range (~25–50 Å) could be reported. This unmodeled density (in gray) blooms around the putative outer membrane region of the protein (Supplementary Fig. 2e). This density appears both on the outside of the beta barrel, and inside, coating the inner lip of

VcPilQ. It remains unclear precisely which residues of VcPilQ are embedded within the outer membrane in situ and whether the membrane spanning thickness is sufficient to fully penetrate the outer membrane (see "Discussion").

**Cysteine pair mutants reversibly inhibit natural transformation and surface piliation.** VcPilQ is thought to mediate DNA uptake during natural transformation in *V. cholerae*[30]. *V. cholerae* can undergo horizontal gene transfer in chitin biofilms, which can promote the spread of antibiotic resistance genes and virulence factors[2]. Blocking DNA uptake by locking the VcPilQ gate with a small molecule could prevent this spread of genetic material. To provide a proof of concept, we designed cysteine pair mutants to reversibly lock the gate with disulfide bonds (Fig. 3a). The structure of VcPilQ was analyzed using the Disulfide by Design 2.0 web tool[72–74] to identify residue pairs with geometries that could support a disulfide, and the top hits were analyzed in UCSF Chimera. The S448C/S453C pair is in the proximal hairpin of the gate, likely cross-linking adjacent VcPilQ monomers (Fig. 3b). The L445C/T493C pair likely cross-links the upper and lower gate hairpins of a single VcPilQ monomer (Fig. 3c). From the design, we hypothesized that the disulfide bonds would be exposed to the extracellular space.

To test if disulfide bonds could form in the gate region under normal culturing conditions, cysteine pair mutant strains were generated (Supplementary Tables 1 and 2) for natural transformation assays. Transformation efficiency was normalized by comparison to the parent strain, which expresses His-tagged VcPilQ (Fig. 3d). The dashed horizontal line marks equivalent efficiency to the parent strain. High levels of the reducing agent dithiothreitol (DTT) were toxic to wild-type *V. cholerae* cells and inhibited natural transformation, so subinhibitory concentrations of DTT were used that allow the His-tagged VcPilQ parent strain to perform natural transformation (Supplementary Fig. 6). Under oxidizing conditions, both cysteine pair mutants demonstrated lower transformation efficiency than the wild-type His-tagged PilQ parent strain (Fig. 3d). The transformation efficiency is not completely ablated in the cysteine pair mutants, which suggests that disulfide bond formation may not be 100% efficient. In the presence of reducing agent, the transformation efficiency of the cysteine pair mutants recovered transformation efficiency similar to the control parent strain (Fig. 3d). This suggests that the two cysteine pair mutants were able to assemble into functional T4aP.

To investigate if the disulfide bonds affect surface piliation, we used strains where the major pilin contains a mutation (*pilA* S67C —aka PilA-Cys) that allows for subsequent labeling with fluorescently conjugated maleimide dyes[31,75]. The *V. cholerae* T4aP used in competence are highly dynamic (much higher than that described for many other pilus systems), such that within a snapshot, very few cells will have surface exposed pili[31]. As a result, deletion of the retraction ATPase *pilT* results in a hyperpiliated phenotype and provides a more sensitive readout for pilus assembly[31,75]. Thus, surface piliation was qualitatively assessed in cysteine pair mutants by fluorescence microscopy under several concentrations of reducing agent using strain backgrounds containing *pilA*-Cys and Δ*pilT* (Supplementary Table 1)[31,75,76]. The cysteine pair mutants exhibited no surface pili under oxidizing conditions (0 mM DTT, Fig. 3e) compared to the parent, which is consistent with pili not being assembled under these conditions. Conversely, piliation was comparable to the parent under reducing conditions (Fig. 3e–g). The S448C/S453C mutant is slightly more recalcitrant to DTT rescue, which may be due to the fact that these cysteines are located further down in the gate region making them less accessible to the reducing agent.

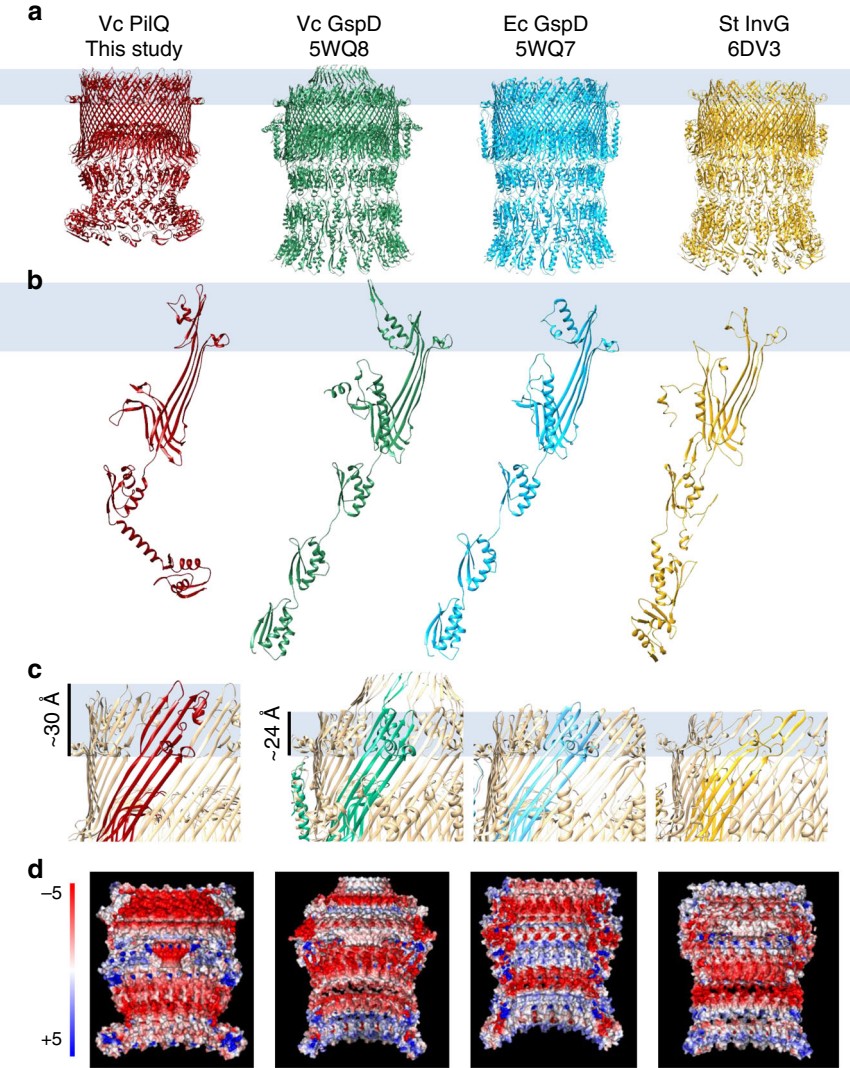

**Fig. 2 Comparison of VcPilQ to T2SS and T3SS secretins.** The structures of VcPilQ (this study, dark red), *V. cholerae* GspD (5WQ8) (green), *E. coli* K12 GspD (5WQ7) (blue), and *S. typhimurium* InvG (6DV3) (yellow) are compared[55, 61]. The putative outer membrane location is depicted as a blue rectangle. The structures are shown as multimers (**a**) or monomers (**b**). **c** The multimer is shown in tan with one subunit colored. The putative outer membrane region is highlighted to show that VcPilQ (left, red) has a ~30 Å membrane spanning distance, while VcGspD, EcGspD, and StInvG have about a 24 Å membrane spanning distance. **d** Adaptive Poisson–Boltzmann solver was used to calculate the electrostatic potential calculation of each secretin in PyMol[142]. The inner cavity of each secretin is shown. The scale varies from −5 (red) to +5 (blue) in units of $K_bT/e_c$.

**Diversity of T4aP secretin domain architecture in Proteobacteria.** Finally, we sought to contextualize VcPilQ within T4aP secretins. T4P function in motility, communication, surface sensing, and DNA uptake[16]. We wondered if the functional diversity of T4aP is reflected in the architecture of secretins. We compared the domain architecture and gene neighborhood of a nonredundant set of 197 representative sequences of T4aP secretins collected from Proteobacteria. We also used phylogenetic inference of the secretin to cluster the sequences into evolutionarily close groups (Fig. 4a). Secretins do not provide a strong phylogenetic signal, which translates into a poorly resolved inference[77]. Next we mapped the domain architecture of the secretins onto the phylogenetic tree (Fig. 4b). In this set of secretins, we found that the dominant domain architecture can be expressed by a variable number of AMIN domains, followed by a N0 (PFAM family STN) domain, a N3 (Secretin_N) domain, and a secretin domain. The number of AMIN domains includes none (27 sequences), one (19 sequences), two (141 sequences), or three (10 sequences) AMIN domain repeats (Supplementary Data 1).

T4aP have been implicated in natural transformation in members of the *Haemophilus*, *Moraxella*, *Neisseria*, *Pseudomonas*, and *Vibrio* genera, but many genera have not been tested[4]. In *Neisseria gonorrhoeae*, the gonococcal pilus (a T4aP) mediates adhesion, twitching motility, and competence, whereas in *P. aeruginosa*, the PAK pilus (a T4aP) is used for adhesion and motility, but not competence[78,79]. In *V. cholerae*, the T4aP used for competence may play a role in adhesion and kin recognition, but does not promote motility[30,37]. We mapped the *Haemophilus*, *Moraxella*, *Neisseria*, *Pseudomonas*, and *Vibrio* genera on our phylogenetic tree and observed that the ability to perform natural competence is not correlated to a particular architecture of T4aP secretin, at least in Proteobacteria (Fig. 4a and Supplementary Data 1).

Finally, we used GeneHood to identify homologs of the proteins located in the genome neighborhood of *pilQ* (Supplementary Data 1). Essentially all of the secretins of the data set were found within the complete *pilMNOPQ* cluster, with a few exceptions. Consistent with literature precedent[80–86], we found that the *pil* cluster is flanked by two genes involved in the Shikimate pathway

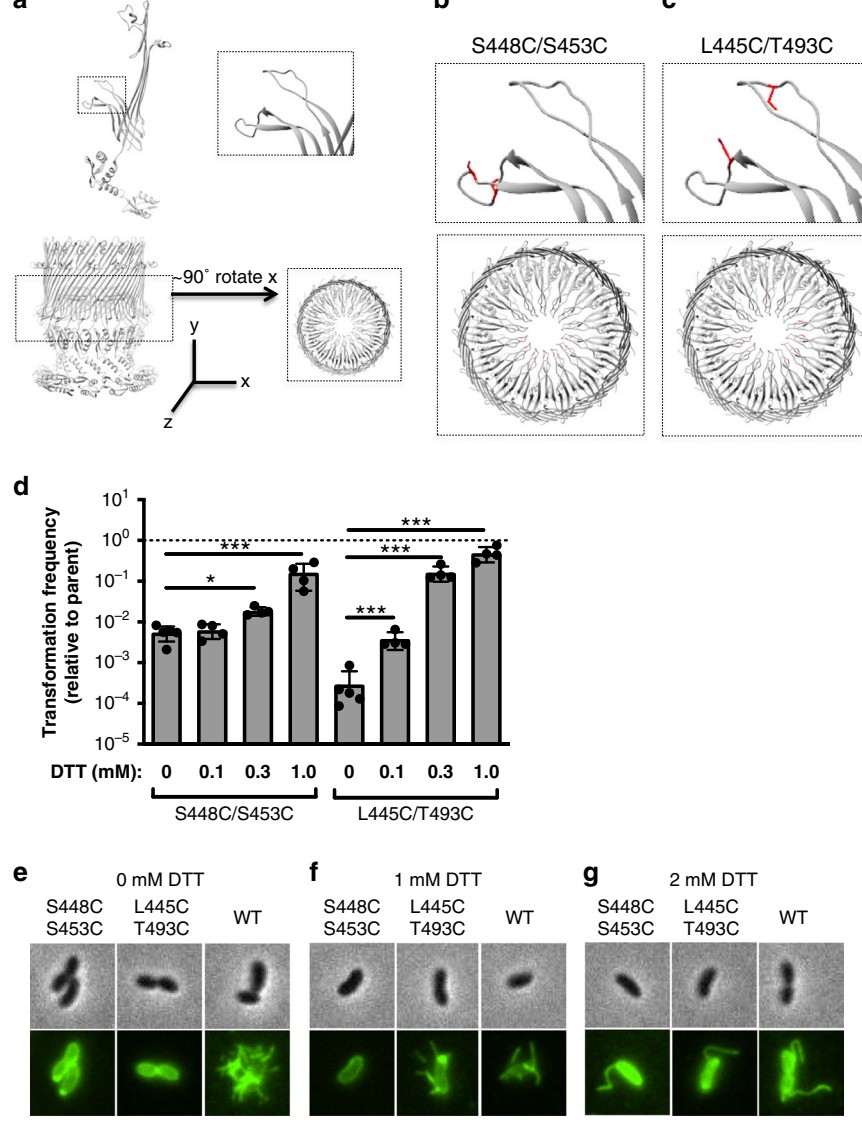

**Fig. 3 Cysteine pair mutants lock the gate, reduce piliation, and reversibly inhibit transformation. a** top: the atomic model of VcPilQ is shown as one chain. The upper and lower beta hairpins of the gate are highlighted in a dashed line box (left) and magnified (right). Bottom: the atomic model of VcPilQ is shown as a multimer and a dashed line rectangle highlights the gate region (left), which is magnified (right) after a 90° rotation about the *x*-axis. The coordinate system is shown. The magnified regions from (**a**) are shown for (**b**) the VcPilQ S448C/S453C mutant, which locks the lower gate and for (**c**) the VcPilQ L445C/T493C mutant, which links the upper and lower gates. In (**b, c**) the residues mutated to cysteine are shown in red. **d** The transformation frequency of two cysteine pair VcPilQ mutants (left: S448C/S453C (Strain TND2169) and right: L445C/T493C (Strain TND2170)) is plotted. Data are normalized to the parental strain that expresses wild-type VcPilQ (Strain TND2140). Natural transformation assays were performed in the presence of varying concentrations of dithiothreitol (DTT) (0–1.0 mM). Data are from at least four independent biological replicates and shown as the mean ± standard deviation. The dashed line indicates the transformation frequency expected if mutants are equivalent to the parental strain expressing wild-type VcPilQ. Statistical comparisons were made by one way ANOVA with Tukey's posttest. *$p < 0.05$, ***$p < 0.001$. The raw data are available in Supplementary Fig. 6. Representative phase contrast (top) and epifluorescence images (bottom) of *V. cholerae* in the hyperpiliated background expressing wild-type (WT) VcPilQ (Strain TND2244), or a cysteine mutant pair (S448C/S453C (Strain TND2242) or L445C/T493C (Strain TND2243)) grown in the presence of **e** 0 mM, **f** 1 mM, or **g** 2 mM dithiothreitol (DTT) prior to labeling with AlexaFluor 488-maleimide to visualize bacterial pili. More than 200 cells were imaged per condition and representative images of cells with pili are shown. In conditions where no cells analyzed exhibited an external pilus, a representative image of a non-piliated cell is shown.

(*aroK* and *aroB*) downstream, and the penicillin-binding proteins 1 A (*mrcA*, *ponA*) which is involved in cell wall formation[87] upstream of it although in a different strand.

## Discussion

Here we present the high-resolution structure of a bacterial T4aP secretin. This protein complex facilitates DNA uptake into diverse bacterial species to aid in their evolution. The *V. cholerae* T4aP is a model system to study natural transformation in bacteria. We observed key differences in the outer membrane region and the periplasmic region among the different members of the secretin family. These differences emphasize the weakness in relying on homology models of evolutionarily related secretins, like the T2SS secretin GspD, to understand VcPilQ. We also discuss the domain architecture of VcPilQ in the context of related T4aP

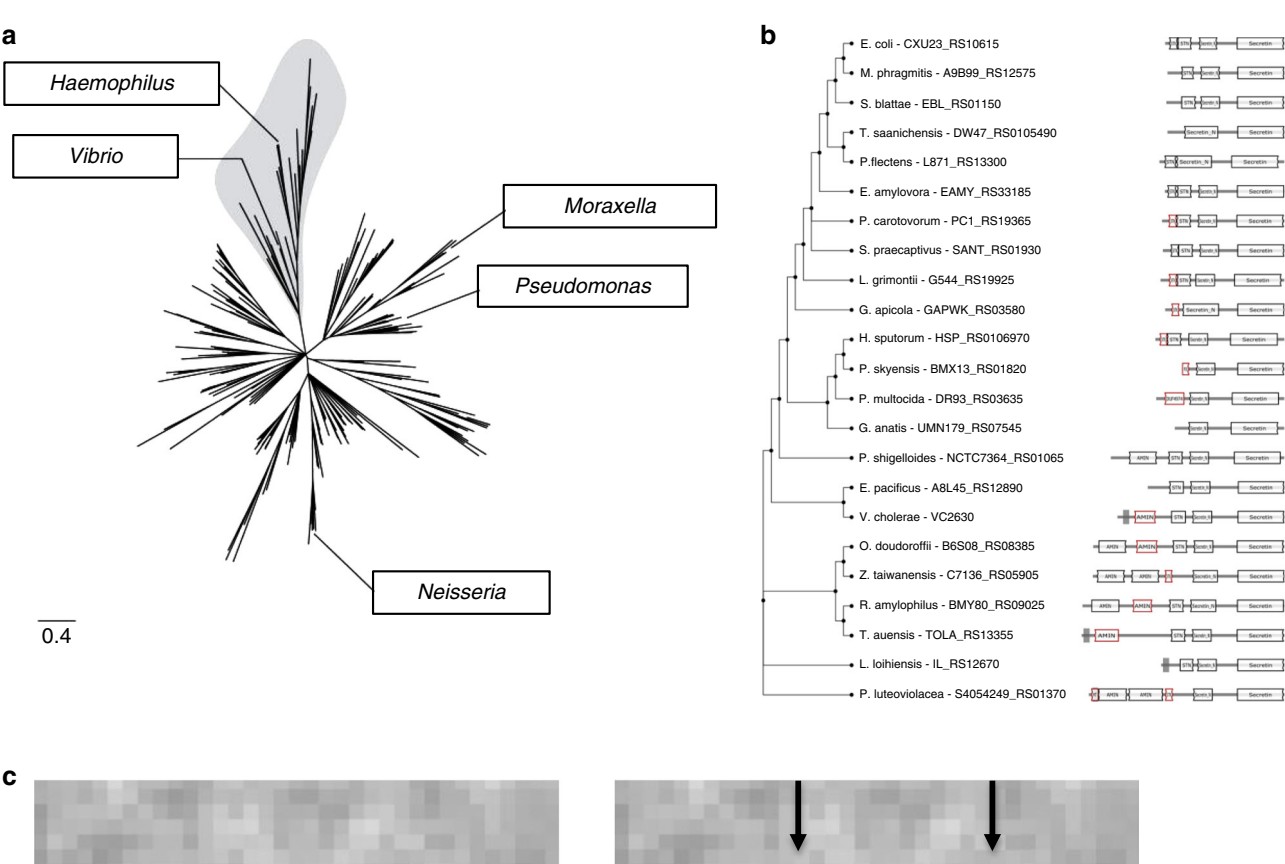

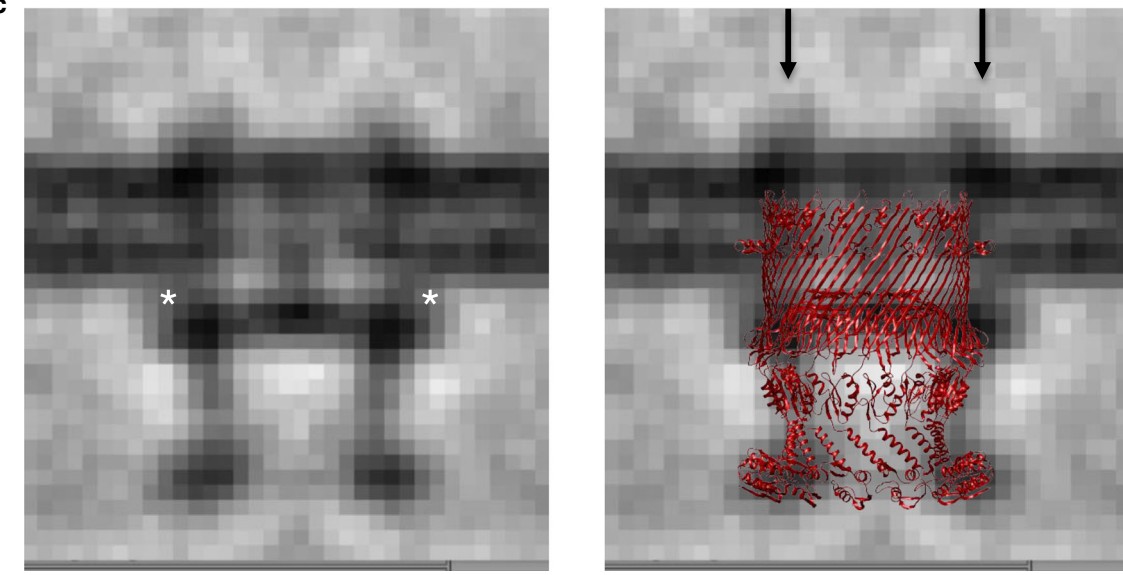

**Fig. 4 The domain architecture diversity of T4aP secretins and a comparison of VcPilQ with the sub-tomogram average of *M. xanthus* type IVa pilus machinery. a** Phylogenetic inference of the secretin domain in a representative set of 197 secretins in Proteobacteria. The shaded region is a fully resolved part of the tree that includes the VcPilQ and is displayed in details in panel (**b**). The locations of representatives of the *Haemophilus, Moraxella, Myxococcus, Neisseria, Pseudomonas*, and *Vibrio* genera are indicated. **b** Detail of the branch that includes the *V. cholerae* (Vc) secretin with domain architecture for each representative sequence. The domains were identified using Comprehensive Domain Visualization Tool (CD-VIST) pipeline by first executing a HMMER search (a hidden Markov model package) (black) and subsequently by a HHblits (HMM–HMM-based lightning-fast iterative sequence search) search (red), while transmembrane regions were predicted using the TMHMM (transmembrane hidden Markov model) package (gray region)[141]. The tree and the gene neighborhood information is available in Supplementary Data 1–3. **c** VcPilQ is compared to the non-piliated sub-tomogram average of the type IVa pilus machinery in the ΔpilP *Myxococcus xanthus* strain (EMD-3249)[49]. In the ΔpilP strain, only PilQ and TsaP are present in the complex. The putative TsaP location is marked with an asterisk in the left image. In the right image, the structure of VcPilQ (dark red) is aligned to the sub-tomogram average. Arrows indicate unidentified extracellular densities.

secretins. We designed cysteine pair mutants to reversibly seal the VcPilQ gate and inhibit natural transformation, which can be used as a tool to further investigate the function of the T4aP machinery in situ. We also compare our structure to previous T4aP sub-tomogram averaging results in *Myxococcus xanthus*

which call in question which parts if any of PilQ extend across the outer membrane. Together, these results elucidate the structure of PilQ and provide a foundation for future mechanistic studies.

In some previous secretin structures, the trans-outer-membrane region appears to be only 2–3 nm thick (Fig. 2c),

which is much thinner than a typical membrane[55,61,71]. In *Thermus thermophilus*, the cryoEM structure of the secretin (TtPilQ) contained a "crown domain" that was unaccounted for by the sequence of TtPilQ[47]. This has left it unclear which regions of secretin molecules are actually embedded in the outer membrane, and whether any residues are exposed to the extracellular surface. Based on the atomic model, we estimate the putative outer membrane region in VcPilQ is at least 3 nm thick and we wondered how this compares to real membranes. The in situ structure of VcPilQ is not available, but an in situ structure of the related *M. xanthus* T4aP (MxT4aP) has been solved by sub-tomogram averaging[49]. We docked VcPilQ into a sub-tomogram average of the MxT4aP secretin (EMD-3249) (Fig. 4c). The gate region of VcPilQ nicely superimposes onto the gate of the *M. xanthus* secretin, which suggests the VcPilQ structure was positioned correctly. However, the putative outer membrane region in VcPilQ does not extend across the entire outer membrane of the MxT4aP sub-tomogram averages. The shortness of the other predicted transmembrane domains of secretins (Fig. 2c) and the two bumps outside the outer membrane in the sub-tomogram average directly above the secretin barrel (arrows in Fig. 4c) call into question whether and how parts of PilQ may penetrate the outer membrane, and if any other accessory proteins are also present in the membrane in situ.

The T4aP machinery extends and retracts the T4aP through the PilQ outer membrane pore[30,88–90]. It remains controversial if the DNA would associate only with the tip of the pilus[31,91] or along the length of the pilus[92,93]. The DNA may accompany the T4P through the secretin pore to enter the periplasm, or an alternative mechanism is possible[4,30,38,94,95]. Several T4aP structures have been solved by cryoEM and reveal pili diameters of 50–80 Å[93,96–98]. In contrast, the inner chamber of VcPilQ varies in diameter from 25 to 108 Å, and docking the *E. coli* T4aP cryoEM structure[98] into VcPilQ reveals steric clashes at the gate (Supplementary Fig. 7). In Yan et al.[55], a glycine in the VcGspD gate (G453) was identified as a putative hinge point to facilitate gate opening, and showed that the G453A mutant trapped the T2SS secretin in a partially open state. In our VcPilQ structure, the inner channel distance between the corresponding glycine (G439) alpha carbons is about 8 nm (Fig. 4a). Thus, we hypothesize that a gate hinge mechanism could accommodate pili up to 7 nm in diameter. In an open state the gate loops could flip up toward the extracellular space to accommodate the pilus. Additionally, the inner surface of VcPilQ is negatively charged (Fig. 2d and Supplementary Fig. 5f, g). In contrast, the inner surface of the T2SS shows alternating negative and positive electrostatic character (Fig. 2d). These differences may relate to the function of VcPilQ in natural competence: the DNA cargo of the T4aP is also negatively charged, so it is possible that this electrostatic repulsion will help the cargo pass through the cavity, rather than getting stuck. By comparison, for T2SS secretins, the charge alternates in the inner cavity, which could reflect its broad scope of cargo.

Our analysis of putative T4aP secretins in Proteobacteria shows a relatively conserved domain architecture (Supplementary Data 1) that is consistent with literature precedent[35,70,99,100]. The model used to identify T4aP secretin sequences was previously validated by the Rocha lab[77], but it could be biased by our current understanding of T4aP gene cluster organization. Perhaps the conservation of the *pilMNOPQ* cluster is linked to a specific domain architecture of secretin. Notably, while the majority of the T4aP secretin sequences contained the N0 (STN), N3 (Secretin_N), and secretin domains, we observed variation in the presence and abundance of AMIN domain repeats (Supplementary Data 1). Looking specifically in the well-resolved branch containing the VcPilQ, we note the gradual loss of AMIN domain over the evolution of secretin (Fig. 4b). Several peptidoglycan-

binding domains are represented in T4P machinery proteins, including the AMIN domain[70] and the LysM domain[101,102], and may play roles in protein localization and stability[103]. The loss of the AMIN domain in some secretins merits further investigation. Thus, we conclude that the variations we observed in domain architecture, in particular the number of AMIN domains in the N-terminus of the secretin, do not determine functional variability, which is expected. Alternatively, if there is a specialization of the secretin sequences toward function it might occur at the amino-acid level scale.

## Methods

**Bacterial strains and culture conditions.** All *V. cholerae* strains were derived from the El Tor strain E7946[104]. Strains were routinely grown in LB Miller broth and agar. Media were supplemented with kanamycin (50 μg/mL), spectinomycin (200 μg/mL), and/or erythromycin (10 μg/mL) as appropriate. All strains were generated by natural transformation as described below in "Natural transformation assays"[64,105]. Mutant constructs were generated by splicing-by-overlap extension PCR to stitch the upstream region of homology (UP arm; amplified with F1/R1), the mutational cargo (MIDDLE arm), and the downstream region of homology (DOWN arm; amplified with F2/R2) together. The full descriptions of the strains are available in Supplementary Table 1 and the primers used are in Supplementary Table 2. The following systems were knocked out of all *V. cholerae* strains used in this study: the MSHA pilus (ΔMSHA::CarbR), the toxin co-regulated pilus (TCP) (ΔTCP::ZeoR), and the cholera toxin (ΔCTX::KanR). Also, all strains used in this study contained mutations to constitutively activate natural competence[31]; the quorum sensing regulatory protein LuxO was knocked out (ΔluxO) and expression of the master regulator of competence, TfoX, was placed under control of the IPTG-inducible Ptac promoter (Ptac-tfoX)[64,104–106]. A deca-histidine tag was added to the N-terminus of PilQ at the native locus to facilitate protein purification. The full genotype of this strain (TND1751) is 10xHis-PilQ, ΔVC1807::SpecR, lacZ::lacIq, comEA-mCherry, ΔluxO, Ptac-tfoX, ΔTCP::ZeoR, ΔMSHA::CarbR, ΔCTX::KanR.

For purification of PilQ, *V. cholerae* expressing His-tagged PilQ (Strain TND1751) were streaked on Luria Broth agar plates and grown overnight at 30 °C. Small cultures were seeded (5 mL) and grown overnight at 30 °C. The next day, 500 mL cultures were seeded with the 5 mL culture. LB broth was supplemented with 20 mM MgCl₂, 10 mM CaCl₂, and 100 μM IPTG to induce expression of TfoX and therefore the T4aP system. The large cultures were grown overnight at 30 °C in beveled flasks. The following day, cultures were spun down (4790 × *g*, 4 °C, 20 min) and the cell paste was weighed, aliquoted, and stored at −80 °C.

For testing the impact of cysteine pair mutants on natural transformation, *V. cholerae* strains were generated where the native copy of PilQ was deleted and the corresponding PilQ allele was expressed at a chromosomally integrated ectopic site under the control of an arabinose-inducible Pbad promoter[107]. The full genotype of the parent strain (TND2140) was ΔlacZ::Pbad-10XHis-PilQ CmR, ΔpilQ::TetR, ΔCTX::KanR, ΔMSHA::CarbR, ΔluxO, ΔTCP::ZeoR, comEA-mCherry, Ptac-tfoX. The cysteine pair mutants were isogenic other than the cysteine mutations introduced into the Pbad-10XHis-PilQ construct, which were PilQ S448C S453C (TND2169) and PilQ L445C T493C (TND2170).

To test the impact of cysteine pair mutants on pilus biogenesis, *V. cholerae* strains were generated akin to those described above, with the exception that the retraction ATPase PilT was deleted[23,30] and the strains contained a cysteine substitution mutation in the major pilin (pilA S67C) that allows for pilus labeling with AlexaFluor 488-maleimide dye[31]. The full genotype of the parent strain (TND2244) was ΔlacZ::Pbad-10XHis-PilQ CmR, ΔpilT::TmR, ΔpilQ::TetR, ΔCTX::KanR, ΔMSHA::CarbR, ΔluxO, ΔTCP::ZeoR, pilA S67C, comEA-mCherry, Ptac-tfoX. The cysteine pair mutants were isogenic other than the cysteine mutations introduced into the Pbad-10XHis-PilQ construct, which were PilQ S448C S453C (TND2242) and PilQ L445C T493C (TND2243).

**Natural transformation assays.** Chitin-independent natural transformation assays were performed[31]. Briefly, competence was induced in late-log phase *V. cholerae* cells as described above. The cells were resuspended in Instant Ocean medium (7 g l⁻¹; Aquarium Systems) and incubated with or without DNA (~500 ng) at 30 °C for 5 h. After the incubation period, cells were shaken with additional LB (1 mL) at 37 °C for 2 h and plated in the presence or absence of antibiotics, and the number of colonies was assessed the following day to calculate transformation frequency. For reactions where strains harbored Pbad-10XHis-PilQ constructs, arabinose was added to a final concentration of 0.2%. Where indicated, DTT was added at the indicated concentrations throughout the assay.

**Competence pilus labeling and microscopy.** Cells were labeled with AlexaFluor 488-maleimide dye and imaged to visualize competence pili[31,75]. Competence was induced in late-log phase *V. cholerae* cells as described earlier in methods. Cells were labeled with AlexaFluor 488-maleimide dye (25 μg/mL) in Instant Ocean with 20 mM MgCl₂ and 10 mM CaCl₂, and then imaged using a Nikon Ti-2 microscope

at ×60 magnification. All strains were grown with arabinose added to a final concentration of 0.2% to induce expression of the Pbad-10XHis-PilQ construct. Where indicated, cells were grown in the presence of the indicated concentration of DTT prior to labeling. More than 200 cells were imaged per condition, and representative images are reported.

**Purification.** Cell pellet (15 g) was resuspended in lysis buffer (50 mM Tris HCl, pH 8, 300 mM NaCl, 1% DDM, 20 mM imidazole) supplemented with lysozyme (40 mg/mL in 50% glycerol/water), DNAse I (4 mg/mL in 50% glycerol/water), and EDTA-free Protease Inhibitor tablet (Roche, 11697498001). Lysis proceeded with stirring at 4 °C for 20 h. Lysate was clarified by ultracentrifugation (Beckman L8-M ultracentrifuge, Rotor Type 45 Ti, $50,000 \times g$, 1 h). The supernatant was mixed with Ni NTA agarose beads (Anatrace, SUPER-NINTA25) and incubated with stirring (4 °C, 8 h). In a gravity column at 4 °C, proteins conjugated to Ni NTA agarose beads were washed (50 mM Tris HCl, pH 8, 300 mM NaCl, 0.05% DDM, 70 mM imidazole), (50 mM Tris HCl, pH 8, 300 mM NaCl, 0.05% DDM, 300 mM imidazole), and eluted (50 mM Tris HCl, pH 8, 300 mM NaCl, 0.05% DDM, 1 M imidazole). Eluant was concentrated to ~1 mg/L (EMD Millipore Amicon Ultra-15, 30 kDa cutoff, UFC903024). Concentrated PilQ (150 μL of ~1 mg/mL protein) was exchanged into Amphipol A8-35 (0.585 mg for a 3:1 ratio, Anatrace, A835) and allowed to incubate at 4 C for 1 h. Excess DDM was removed using Bio-Beads SM2 (Bio-Rad, 1523920) by incubating overnight at 4 °C. The protein was concentrated. Protein was analyzed on Bio-Rad Any kD™ Mini-PROTEAN+ TGX Stain-Free™ Protein Gels (Bio-Rad, 4568126) by stain-free exposure, Coomassie staining, or western blot with 6x-His Tag Monoclonal Antibody (HIS.H8), HRP (Invitrogen, MA1-21315-HRP).

**Electron microscopy.** For cryoEM, Quantifoil R2/2 300 Mesh grids (EMS, Q33100CR2) were glow discharged (Pelco EasiGlow, 20 mA, 60 s). PilQ in amphipol (3 μL of ~0.8 mg/L) was frozen on a Mark IV Vitrobot (FEI, 20 °C, 100% relative humidity, blot force −6, blot time 4 s). Micrographs were collected on a 300 kV Titan Krios microscope (FEI) with energy filter (Gatan) and equipped with a K3 direct electron detector (Gatan). Data were collected using Serial EM software with a pixel size of 1.104 Å (×81,000 magnification) and a defocus range from −1.0 to −3.0 μm[108]. A fluence of 19.8 electrons/pixel/second was used with a 3.7 s exposure time to collect 60 e−/ Å².

**Image processing.** The cryoEM image processing workflow is summarized in Supplementary Fig. 3. MotionCor2 was used for motion correction and dose weighting of 3808 movies[109]. Contrast transfer function (CTF) correction was used to evaluate micrograph quality[110]. CryoSPARC blob picking on 2510 micrographs yielded 3,100,353 potential particles[111]. After inspection, the 252,319 particles were analyzed by several rounds of 2D classification and 3D classification to yield 100,543 particles. These particles were moved to Relion using the UCSF PyEM package script (https://github.com/asarnow/pyem/)[112]. In Relion, several rounds of 3D refinement, polishing, and CTF refinement were used[113–115]. ResMap was used to calculate local resolution[68].

**Model building and refinement.** The initial model (residues 230–571) was auto-built using Buccaneer[116]. Subsequent building and model adjustments were performed by hand using COOT[117]. A homology model of the N0 domain (residues 160–229) was created using I-TASSER and manually docked using COOT[118–120]. Coulombic potential density for residues 1–159 was not observed. The model was refined in PHENIX version 1.16-dev3549 using phenix.real_space_refine with the resolution set to 3 Å[121]. NCS constraints were applied for the 14 subunits and were automatically detected and refined. Automatically determined secondary structure restraints, rotamer restraints, and Ramachandran restraints were applied as well. The quality of the model was evaluated using EMRinger[122] and Molprobity[123] (Table 1).

**CryoEM structure analysis.** The structure of VcPilQ was compared to the *V. cholerae* T2SS secretin GspD (PDB 5WQ8), the *E. coli* T2SS GspD (PDB 5WQ7), and the *S. typhimurium* T3SS InvG (PDB 6DV3)[55,124]. UCSF Chimera Match-Maker was used to calculate the RMSD between the C-alpha carbons of each pair[125,126].

The Positioning of Proteins in Membrane web server was used to predict the location of the transmembrane region of VcPilQ[127]. Using UCSF Chimera, the structure of VcPilQ was docked into sub-tomogram averages of the non-piliated MxT4aP in *M. xanthus* ΔpilQ (EMD-3249), the non-piliated MxT4aP in *M. xanthus* ΔpilB (EMD-3260), and the piliated MxT4aP in wild-type *M. xanthus* (EMD-3247)[49,125]. At a high cryoEM density threshold, the lipid bilayer is clearly observed as two leaflets in the MxT4aP sub-tomogram averages, so there was little doubt where the hydrophobic region of VcPilQ should be placed in the sub-tomogram average. The bilayer produces stronger features in the MxT4aP sub-tomogram averages than the protein, so a lower cryoEM density threshold is required to visualize the full MxT4aP machinery. Here we focus on the ΔpilP MxT4aP sub-tomogram average (EMD-3249) because in this mutant only PilQ and TsaP localize correctly, and are therefore the only proteins likely to be present in the sub-tomogram average (Fig. 4c)[49]. The putative location of TsaP suggested in

Chang et al.[49] is marked with an asterisk in Fig. 4c. In *M. xanthus*, TsaP is a peptidoglycan-binding protein[102]. In *V. cholerae*, the TsaP homolog LysM has not been implicated in T4aP function.

**Unmodeled density analysis.** The UCSF Chimera command molmap was used to generate a synthetic three-angstrom resolution cryoEM density map based on the VcPilQ atomic model (VcPilQ.pdb) (command: molmap <VcPilQ.pdb model number> 3)[125]. The resulting density map (molmap_VcPilQ.mrc) was resampled to match the pixel size and box size of the VcPilQ cryoEM density map (the output from Relion Refine3D, cryoEM_PilQ.mrc)[114]. Resampling was performed in UCSF Chimera (command: vop resample <molmap_VcPilQ.mrc model number> onGrid <cryoEM_PilQ.mrc model number>), which yielded a synthetic density map based only on the atoms in the atomic model (molmap_VcPilQ_resampled.mrc).

The Relion command relion_mask_create was used to generate an inverted mask based on the synthetic density map (command: relion_mask_create --i molmap_VcPilQ_resampled.mrc --o mask_molmap_VcPilQ_resampled.mrc --ini_threshold 0.013 –invert). The resulting mask (mask_molmap_VcPilQ_resampled.mrc) has zeros where atoms are modeled and ones everywhere else (unmodeled space). To remove the parts of the empirical cryoEM map where atoms are modeled, the inverted mask is multiplied by the empirical cryoEM map (command: relion_image_handler --i cryoEM_PilQ.mrc --o cryoEM_PilQ_unmodeled_density.mrc --multiply mask_molmap_VcPilQ_resampled.mrc). The result is a cryoEM density volume representing unmodeled density (cryoEM_PilQ_unmodeled_density.mrc).

**Mass spectrometry.** After running a Bio-Rad Stain-Free gel and performing a Coomassie staining, the band of interest was excised with a clean razor blade. The gel piece was destained with ammonium bicarbonate and reduced with DTT (50 ° C, 30 min). Next the sample was alkylated with iodoacetamide (room temperature, dark, 20 min). The gel pieces were then dehydrated. Trypsin was used to digest the protein in the gel (37 °C, overnight). Peptides were extracted from the gel matrix, dried, and desalted with a zip tip.

The in-gel-digested samples were subjected to LC-MS/MS analysis on a nanoflow LC system, EASY-nLC 1200 (Thermo Fisher Scientific), coupled to a QExactive HF Orbitrap mass spectrometer (Thermo Fisher Scientific, Bremen, Germany) equipped with a Nanospray Flex ion source.

Samples were directly loaded onto a C18 Aurora series column (Ion Opticks, Parkville, Australia). The 25 cm × 50 μm ID column (1.6 μm) was heated to 45 °C. The peptides were separated with a 60 min gradient at a flow rate of 350 nL/min. The gradient was as follows: 2–6% Solvent B (3.5 min), 6–25% B (42.5 min), and 25-40% B (14.5 min), to 100% B (1 min) and 100% B (12 min). Solvent A consisted of 97.8% water, 2% acetonitrile, and 0.2% formic acid and solvent B consisted of 19.8% water, 80% acetonitrile, and 0.2% formic acid.

The QExactive HF Orbitrap was operated in data-dependent mode. Spray voltage was set to 1.8 kV, S-lens RF level at 50, and heated capillary at 275 °C. Full scan resolution was set to 60,000 at $m/z$ 200. Full scan target was $3 \times 10^6$ with a maximum injection time of 15 ms (profile mode). Mass range was set to 300–1650 $m/z$. For data-dependent MS2 scans the loop count was 12, target value was set at $1 \times 10^5$, and intensity threshold was kept at $1 \times 10^5$. Isolation width was set at 1.2 $m/z$ and a fixed first mass of 100 was used. Normalized collision energy was set at 28. Peptide match was set to off, and isotope exclusion was on. Ms2 data were collected in centroid mode.

Raw data were analyzed using MaxQuant (v. 1.6.5.0)[128,129]. Spectra were searched against UniProt *V. cholerae* entries (3784 sequences) and a contaminant protein database (246 sequences). Trypsin was specified as the digestion enzyme and up to two missed cleavages were allowed. Precursor mass tolerance was 4.5 ppm after recalibration and fragment mass tolerance was 20 ppm. Variable modifications included oxidation of methionine and protein N-terminal acetylation. Carbamidomethylation of cysteine was specified as a fixed modification. A decoy database was used to set score thresholds to ensure a 1% false discovery rate at the protein and peptide level. Protein abundances were estimated using iBAQ and the fractional abundance was calculated as the protein abundance divided by the sum of all non-contaminant protein abundances[130].

**Bioinformatics resources and software.** Sequences were selected in the MiST3 database[131] as of June 2020. The secretin protein domain model was taken from the PFAM database[132] and specific models for individual type four filamentous family from the Rocha Lab[77]. Matches to the protein domains were found using HMMER 3.1b2[133]. Data manipulation was executed by custom scripts written in Typescript and available at https://gitlab.com/jensenlab/seccomp. To decrease the redundancy of the unaligned secretin sequences we used CD-HIT v4.6[134]. Similarity scores were calculated with BLAST v2.7.1+[135]. To further filter out divergent sequences, we used dyno cluster (https://gitlab.com/jensenlab/dyno-cluster) to parse the blastp results and generate connected graphs. Sequences were aligned using L-INS-I from the software package MAFFT v.7305b[136]. We used Jalview[137] to manually inspect the multiple sequence alignment. We used the TREND[138] domain pipeline for initial explorations of the data set. Phylogenetic inferences were built using RAxML v8.2.1059[139]. To collapse branches with low support in phylogenetic trees, we used TreeCollapseCL4 v3.0[140]. We used CD-VIST for detailed domain architecture

identification and visualization[141]. The gene neighborhood data set and images were generated by GeneHood command line application v0.2.8-1 (https://npmjs.org/package/genehood-cli) and visualization together with homolog assignment using GeneHood viewer v0.16.0 (https://genehood.io).

**Secretin sequence selection and analysis**. We selected all 198,900 genes from the MiST3 database to which the gene product contained at least a single match to the Pfam domain model of secretin from Proteobacteria genomes. MiST3 contains pre-computed domain architecture prediction against Pfam 31 and taxonomy ranks from the National Center for Biotechnology Information. We used hhpress from HMMER to generate a hidden Markov model database from the secretin models defined by the Rocha Lab[77]. We selected 56,941 sequences had the highest score against the T4aP model and above $1E-40$ $e$ value. Next, we trimmed the region of the sequences matching the boundaries of the secretin domain model from Pfam 33. We used CD-HIT to decrease the redundancy of the data set at 65% identity to a total of 386 representative trimmed sequences and recollect the full-length sequences of these representatives. We used dyno cluster with $1E-110$ threshold to select over 50% of the sequences in the largest connected subgraph. We aligned the 203 full-length sequences using L-INS-I. We eliminated six sequences that open large gaps in the alignment. With the final data set of 197 sequences, we generated a phylogeny inference using RAxML with "-m PROTGAMMAILG -p 1234555 -x 9876545 -f a -N 200" parameters. We used TreeCollapse to collapse nodes with <50% bootstrap support. The alignment and the phylogenetic tree can be found in Supplementary Data 2. Also, we submitted the sequences to CD-VIST to detail domain architecture analysis with default parameters but skipping RPSBLAST step. Finally, we ran genehood-cli to fetch six genes up and downstream of each gene identifier in the alignment. We use GeneHood viewer to visualize the gene neighborhoods mapped to the phylogeny. GeneHood uses BLAST all vs. all similarity scores as edges and each gene of the data set as vertices in graphs to search for homologs among the displayed genes. The software filters edges according to a selected threshold for each gene and performs a breadth-first search to find all vertices that belong to the connected subgraph of the selected gene. All vertices of the selected subgraph are marked with the same color as the selected gene. Using this feature, we marked the homologs of each *Vibrio* gene in the data set. The thresholds used can be accessed by loading the Supplementary Data 3 to genehood.io.

**Reporting summary**. Further information on research design is available in the Nature Research Reporting Summary linked to this article.

## Data availability
Data supporting the findings of this paper are available from the corresponding author upon reasonable request. The cryoEM reconstruction and model have been deposited in the Electron Microscopy Data Bank (https://www.ebi.ac.uk/pdbe/entry/emdb/EMD-21559) and the Protein Data Bank (PDB 6W6M).

## Code availability
The scripts underlying the bioinformatics analysis are available at https://gitlab.com/jensenlab/seccomp. The alignment and the phylogenetic tree are available in Supplementary Data 2, while the thresholds used in the genehood.io analysis are in Supplementary Data 3. Other data are available from the corresponding author upon reasonable request.

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

## Acknowledgements

Cryo Electron microscopy was performed in the Beckman Institute Resource Center for Transmission Electron Microscopy at Caltech. Dr. Songye Chen and Dr. Andrey Malyutin assisted with data collection. Dr. Spiros D. Garbis, Dr. Annie Moradian, Dr. Michael Sweredoski, and Dr. Brett Lomenick at the Caltech Proteome Exploration Laboratory (PEL) performed and analyzed mass spectrometry results. Dr. Naima Sharaf, Jeffery Lai, and Prof. Doug Rees provided invaluable advice on membrane protein biochemistry and instrumentation. Jane Ding and Welison Floriano provided computational support. Dr. Debnath Ghosal, Dr. Mohammed Kaplan, Dr. Catherine Oikonomou, Dr. Lauren Ann Metskas, Dr. Christopher Barnes, Claudia Jette, and Andrew Schacht provided feedback and advice. This work was supported in part by National Institutes of Health grant AI127401 to G.J.J. and National Institutes of Health grant R35GM128674 to A.B.D.

## Author contributions

S.J.W. conceptualized the project, expressed and purified the protein, prepared samples for cryoEM, collected cryoEM data, processed cryoEM data, assisted in atomic model building, interpreted results, designed figures, and wrote the paper. D.R.O. designed research, performed sequence analyses, wrote software, collected sequence data, interpreted results, designed figures, and contributed text to the paper. M.H.S. purified protein, assisted with cryoEM sample prep and data collection, built the atomic model, interpreted results, and provided feedback on the paper. T.N.D. engineered the *V. cholerae* constructs, performed microbial assays, and interpreted results. A.B.D. conceptualized the project, obtained funding, engineered the *V. cholerae* constructs, performed microbial assays, interpreted results, designed figures, and provided feedback on the paper. G.J.J. conceptualized the project, obtained funding, interpreted results, and provided feedback on the paper.

## Competing interests

The authors declare no competing interests.
