## [Peer Review File · Nature Communications]

Reviewers' comments:

Reviewer #1 (Remarks to the Author):

The manuscript by Sara. et al provides a high-resolution structure of the T4SS secretin PilQ. The higher resolution enabled more detailed structural information than previous works, and showed new features different from other secretins. The major part of the complex is still remained similar to other secretins. Careful comparisons were made to compare the PilQ with other secretins, and possible mechanism of the Pili transfer is speculated. The structure is interested, and provides some new insights to the secretin.

Comments and suggestions

- 1) line 32, "suggesting VcPilQ as new drug target" . This conclusion was drawn based on the mutation in the gate region of PilQ. However, no drugs can make such a mutation. So I don't think this work can make such suggestion. And also in line 30, "We prove that it is possible to reduce pilus biogenesis and natural transformation by sealing the gate", this is obvious and well known before this work.
- 2) line 240, the unmodeled densities were used to measure the thickness of the membrane. This might be problematic, because these densities are from amphipol rather than real membrane. And choosing a proper threshold is also difficult. A better way might be to compare the in situ thickness between PilQ and T2SS or T3SS secretin by cryoET. More importantly, if the thicker membrane is true, why does PilQ take the thicker outer membrane region? Related to lack of a S domain binding to pilotin?
- 3) line 297, failure of homology modeling is meaningless to validate the different features of PilQ from other secretins.

Reviewer #2 (Remarks to the Author):

The manuscript by Weaver et al describes a cryoEM structure of the *Vibrio cholerae* type IV competence pilus secretin VcPilQ to ~ 2.7 Å resolution. Although several high resolution structures have already obtained for the related T2SS and T3SS secretins the resolution of the cryo-EM map is the highest obtained so far for a type IV competence pilus secretin and unravels a C14 symmetry and distinct molecular details of the different domains of VcPilQ and sheds light onto the electrostatic characteristics of the inner surfaces. The *Vibrio cholerae* type IV competence pilus secretin has four conserved domains, AMIN, N0, N3 and the secretin domain of which the N0, N3 and the secretin domain are resolved in the cryoEM map. The structure of the AMIN domain and the structure of a region following the AMIN domain were not resolved. A novel helical coil was identified between the N0 and N3 domain. Moreover the putative outer membrane region formed by the secretin amphipathic helix lip (AHL) and the beta strand of the beta lip was found to be thicker than those of T2SS secretins. On the other hand, functional analyses to interrogate the mechanism of PilQ are less conclusive. In particular, the transformation and piliation analyses of locked PilQ mutants are not convincing. Moreover, the conclusion that the gate must open for pilus biogenesis is trivial because considering the dimensions of the competence pilus gate opening is a prerequisite to extrude the type IV competence pilus.

Overall the work appears well done and paper is clearly and concisely written, but there are still some issues that need to be addressed before the manuscript will be acceptable for publication.

Page 1, line 27 -28: The structural analyses of the PilQ mutants does not unravell the mechanism of PilQ. That the gate has to open to extrude the type IV competence pilus is evident from the dimensions of the type IV competence pilus and the dimensions of the VcPilQ gate. However the structural analyses do not shed any light onto the mechanism of gate closing and opening, on the signals required, how are these signal are transferred, which other components are involved in signal transfer and how does the secretin interact with its substrates such as the competence pilus or DNA?.

Page 7, line 185: The assignment of the AMIN domain to the hazy density present in the 2D classification in Figure 1B needs to be solidified. The same holds true for the residues 126 – 159 following the AMIN domain. Deletion derivatives devoid of the AMIN domain and/or devoid of the region following the AMIN domain in VcPilQ have to be generated and subject to structural analyses.

Page 7, line 195: The authors suggest that the AMIN domains are probably not regularly arranged in situ. The structural data presented do not provide any evidence for this conclusion. To get insights into the in situ arrangement of VcPilQ the authors should analyze the in situ arrangement by cryo tomography of *V. cholerae* cells producing the His-tagged VcPilQ.

Page 8, line 212: The gate of the secretin has the most narrow inner diameter of the PilQ channel. Information with respect to the charge of the inner surface of the gate should be stated in the text.

Page 10, line 261: There is no figure 4C. How was the detergent belt around the putative outer membrane region determined?

Page 11, line 280 – 282: Why do the *V. cholerae* mutants which are blocked in DNA uptake by locking the gate of the PilQ channel only exhibit a reduced natural transformation phenotype but are not completely defect in natural transformation?

Page 11, line 282 -283: Statistics on piliation is missing. How many wt and mutant cells were piliated and how many pili were detected per cell?

Reviewer #3 (Remarks to the Author):

The key result from this paper is the determination of the structure of PilQ from *Vibrio*, which is achieved (for the majority of the protein) at a resolution sufficient to build an atomic model. There are some interesting differences between the T2SS and T3SS counterparts described. Some cysteine mutagenesis is also carried out in the gate region with experimental analysis of the mutants.

Major points:

1. The authors describe the high-resolution structure of the PilQ secretin from *Vibrio*, which they describe as “a Type IV competence pilus secretin”. This is confusing because it is not clear if this is different to the PilQ Type IV secretins in numerous other systems that are not specifically described as “competence secretins”. If there are genuine differences in the type IV pilus systems (competence or not) this would be interesting to highlight and clarity should be provided throughout the entire manuscript e.g. L83-88, L210-219, L249, L378-388, and all figures where comparisons are made. However, if there are no significant differences, the authors should clearly state this from the outset in order to clarify the nomenclature. In attempting to navigate this issue, this author came across a recent Review (Piepenbrink, K. H; 2019; <https://doi.org/10.3389/fmolb.2019.00001>) that describes *Vibrio* as expressing type IV pili (and being naturally competent like many other gram negative

bacteria with type IV pili), whereas Gram positive bacteria express competence pili, which are something different altogether - they do not have PilQ.

2. Related to point 1, on L382, the authors go on to say that the "Vibrio Type IV competence pilus is in fact a Type IVa pilus". This argument is used to justify docking their new Vibrio "competence secretin" PilQ into their previously determined density map for PilQ from the "Type IVa pilus" from Myxococcus. Regardless of the confusion in nomenclature, this simple analysis takes up ~40 lines of text with the outcome being that PilQ may not penetrate the outer membrane fully. This is not written in a way that conveys a particularly exciting result and should be removed or made considerably more succinct. The implications of this should be spelt out more clearly.

3. In their model building, the authors use I-TASSER to build a homology model for the less well resolved NO domain which they then dock into their map. The model is found to be similar to previous structures, which gives confidence that it is correct - but it is somewhat at odds with the considerable words (L299-311) and an entire Figure (Fig. S7) that the authors dedicate to describing how homology modelling is generally insufficient to predict structure. Therefore it seems that Fig. S7 and the accompanying text does not add anything to support the manuscript.

4. The authors show that the introduction of disulphide bonds in the gate region impairs natural transformation and piliation. Being as this would mean that the gate cannot move or open, this is not really surprising. Likewise, the section title "Cysteine mutants indicate gate must open for pilus biogenesis and natural transformation" is somewhat obvious. It's hard to see how a drug would be able to induce such a change (L32). The manuscript would be stronger if the authors could provide more compelling evidence for how they could specifically inhibit gate movement.

5. Why were the cysteine pair mutants made in a PilT deletion strain and not wild-type? As PilT is the retraction ATPase, wouldn't this mean that there would be an over-inflated number of pili assembled on the cell surface? Is this taken into account?

6. The authors assess the level of piliation in their cysteine pair mutants. Fluorescent labelling and light microscopy imaging doesn't seem particularly accurate for determining the amount of pili and it's difficult to assess the result. Readers also need to refer to other papers to follow the methodology. How many cells were tested and how many pili? A higher resolution imaging technique such as electron microscopy should be used where the pili can be observed directly.

7. Related to the above, is there evidence to demonstrate that cysteine mutant PilQ can still assemble in the membranes at wild-type levels?

8. What is the evidence that the pilus and DNA actually fill the diameter of the secretin channel at the same time? There is a lot of discussion about this point but the evidence for it is not clearly described.

9. There is a vast amount of text (L378-416; some 38 lines) about how the PilQ structure could accommodate a pilus. Being as this is the role of PilQ, it's not particularly surprising and should be written more succinctly.

10. Many of the figures should be combined to generate a more succinct manuscript. E.g. Fig. 2 and 4 do need to stand-alone and Fig. 5 and 6 are for the same experiment. Fig. 7 should be supplementary.

Minor points:

1. On L65, the authors describe how chitin on the exoskeletons of crustaceans induces expression of the machinery. How this links to human disease is not made very clear.

2. An AMIN domain is not explained. As the flexibility of these domains is used to justify a lack of observed density in the cryoEM maps (L194), it should be properly described.
3. The authors mention and show in Fig. 2 work from Koo et al and D'Imprima et al, but these could not be found in the bibliography.
4. The description of cryoEM sample preparation and imaging conditions adds an unnecessary 20 lines to the Results section (L159-179). It's already described in Methods and Fig. S4.
5. Is the reference to Fig. S8 on L206 correct?
6. Why are there so many gels in Fig. S1? They appear to show the same thing.

1 **Response to Reviews for Weaver et al 2020:**

2 **KEY:**

3 *Reviewer comments: Arial and italics*

4 Our reply: Arial

**Reviewers' comments:**

**Reviewer #1 (Remarks to the Author):**

*The manuscript by Sara. et al provides a high-resolution structure of the T4SS secretin PilQ.*
*The higher resolution enabled more detailed structural information than previous works, and*
*showed new features different from other secretins. The major part of the complex is still*
*remained similar to other secretins. Careful comparisons were made to compare the PilQ with*
*other secretins, and possible mechanism of the Pili transfer is speculated. The structure is*
*interested, and provides some new insights to the secretin.*

**Comments and suggestions**

*1) line 32, "suggesting VcPilQ as new drug target" . This conclusion was drawn based on the*
*mutation in the gate region of PilQ. However, no drugs can make such a mutation. So I don't*
*think this work can make such suggestion.*

We agree that a small molecule drug would not mutate the protein. We imagined a potential
drug that could bind the PilQ gate and somehow disrupt its normal function, just as we showed
disulfide bonding disrupts its function.

*And also in line 30, "We prove that it is possible to reduce pilus biogenesis and natural*
*transformation by sealing the gate", this is obvious and well known before this work.*

We have revised the text to emphasize that our work confirms and supports existing literature
on this point.

*2) line 240, the unmodeled densities were used to measure the thickness of the membrane.*
*This might be problematic, because these densities are from amphipol rather than real*
*membrane. And choosing a proper threshold is also difficult.*

We agree, and we have now revised the text to further clarify/highlight these concerns, but the
major point we want to call attention to is that the previously published secretin structures vary
in the thickness of their putative transmembrane domains, and all are substantially smaller
than expected for real membranes. Thus something really interesting is happening here, and it
warrants attention: none of the single particle cryoEM structures of secretins (including our
PilQ structure) depict transmembrane region distances that agree with the thickness of real cell
membranes, so if and/or how they fully cross the membrane is in question!

Because we agree with the Reviewer about the uncertainty in choosing the proper cryoEM
density threshold, we estimated the transmembrane domain thickness in two ways. First we
measured the distance on the atomic model (where the hydrophobic residues are). Second,
we inspected the unmodeled (amphipol) density and measured its thickness at different
cryoEM thresholds. We have now added a longer section to the Methods explicitly describing
the unmodeled density protocol.

By the way, two Type II Secretion System secretins have been solved in amphipol: the *E. coli*
EPEC GspD (PDB 5W68) and the *P. aeruginosa* XcpQ (PDB 5WLN)(Hay et al. 2018; Hay,
Belousoff, and Lithgow 2017). Comparing the *E. coli* EPEC GspD in amphipol (MAGENTA PDB
5W68)(Hay, Belousoff, and Lithgow 2017) with the *E. coli* K12 GspD (GREEN PDB 5WQ7)(Yan
et al. 2017) in LDAO detergent reveals an essentially identical fold in the outer membrane:

This suggest the amphipol provides an environment quite similar to detergent for the T2SS
GspD.

A better way might be to compare the *in situ* thickness between PilQ and T2SS or T3SS
secretin by cryoET.

We have revised the manuscript to directly reference Ghosal et al. 2019, where the *in situ*
structure of *Legionella pneumophila* T2SS secretin is compared to the *V. cholerae* T2SS
secretin EpsD and the *E. coli* T2SS secretin GspD (below)(Ghosal et al. 2019; Yan et al.
2017):

Extended Data Fig. 3 | Position of the T2SS secretin with respect to the OM. (a, b) Atomic models of the *V. cholerae* (PDB ID: 5WQ8) and *E. coli* (PDB ID: 5WQ7) T2SS secretins superimposed on our subtomogram average based on the position of the gate. (c) Positions of the OM on these structures as suggested in earlier publications^{13,15}. The widths of the suggested OM spanning regions were only ~1.8 nm, but real membranes are known to be 5-7 nm wide. In all reported atomic models, the secretin channel is suggested to extend beyond the OM^{13,15}. However, when we overlaid the secretin atomic models on our subtomogram average, it only reached through the inner leaflet of the OM. Scale bars, 10 nm. (d) Tomographic slices of mutant *L. pneumophila* cells lacking all major and minor pilins ($\Delta IspGHJK$). Showing representative individual T2SS particles. No pseudopilus or lower-periplasmic ring is visible. A similar result was obtained when we examined a *L. pneumophila* $\Delta IspGHJK$ mutant. Scale bar, 10 nm (d). For each strain, number of tomograms recorded and number of particles found are listed in the SI Table-1.

*More importantly, if the thicker membrane is true, why does PilQ take the thicker outer*
*membrane region? Related to lack of a S domain binding to pilotin?*

We would like to clarify that we don't think the *Vibrio cholerae* outer membrane is thicker than
the outer membrane of the other bacteria with published secretin structures (*E. coli*, *P.*
*aeruginosa*, etc). We aim to point out that none of the published secretin structures contain a

transmembrane region that matches the width of real bacterial outer membranes, and we have
 now revised the text in hopes of making that point clearer.

Regarding whether or not there is a pilotin protein in our system, we note that VC1612 has
 been implicated as a potential pilotin (Metzger and Blokesch 2014). VC1612 expression is
 increased upon exposure to chitin or induction of TfoX transcription factor (Meibom et al. 2005;
 2004). VC1612 is similar to the *Pseudomonas aeruginosa* pilotin PilF (Metzger and Blokesch
 2014) and to the *Neisseria gonorrhoeae* pilotin PilW (PilF) (Seitz and Blokesch 2013). We did
 not see density in our structure, however, that we could attribute to a pilotin protein.

Regarding whether the presence of a pilotin protein would change the thickness of the secretin
 transmembrane domain, see below the results from Howard et al. 2019 where the structure of
 a T2SS secretin from a pilotin-dependent secretin (*Vibrio vulnificus* EpsD) is compared to a
 pilotin-independent secretin (*Aeromonas hydrophila* ExeD)(Howard et al. 2019). We have
 included Figure 5 from that paper to demonstrate that the outer membrane thickness is
 approximately constant, regardless of the presence or absence of the pilotin:

Fig 5. Domain arrangements of the single monomers of ExeD and EpsD and alignment of secretin S-domains. (A) Single monomers of ExeD and EpsD are shown as they would be oriented to the plane of the outer membrane. The S-domains are indicated in red. The outer membrane (OM) is indicated as a yellow bar. In both secretin structures, domain N1 was mostly modeled from X-ray crystal structures. (B) Multiple sequence alignment of secretin S-domains from species *V. cholerae* (5WQ8), *V. vulnificus* (this work), *A. hydrophila* (this work), *E. coli* ETEC (5ZDH), *E. coli* EPEC (5W68), *E. coli* K-12 (5WQ7) and *P. aeruginosa* (5WLN). Amino acids that comprise α -12 are boxed in red. The last residue observed in each cryo-EM structure is boxed in orange.

<https://doi.org/10.1371/journal.ppat.1007731.g005>

 3) line 297, failure of homology modeling is meaningless to validate the different features of
 PilQ from other secretins.

We agree and have modified the manuscript and figures.

**Reviewer #2 (Remarks to the Author):**

*The manuscript by Weaver et al describes a cryoEM structure of the Vibrio cholerae type IV*
*competence pilus secretin VcPilQ to ~2.7 Å resolution. Although several high resolution*
*structures have already obtained for the related T2SS and T3SS secretins the resolution of the*
*cryo-EM map is the highest obtained so far for a type IV competence pilus secretin and*
*unravels a C14 symmetry and distinct molecular details of the different domains of VcPilQ and*
*sheds light onto the electrostatic characteristics of the inner surfaces. The Vibrio cholerae type*
*IV competence pilus secretin has four conserved domains, AMIN, N0, N3 and the secretin*
*domain of which the N0, N3 and the secretin domain are resolved in the cryoEM map. The*
*structure of the AMIN domain and the structure of a region following the AMIN domain were*
*not resolved. A novel helical coil was identified between the N0 and N3 domain. Moreover the*
*putative outer membrane region formed by the secretin amphipathic helix lip*
*(AHL) and the beta strand of the beta lip was found to be thicker than those of T2SS secretins.*
*On the other hand, functional analyses to interrogate the mechanism of PilQ are less*
*conclusive. In particular, the transformation and piliation analyses of locked PilQ mutants are*
*not convincing. Moreover, the conclusion that the gate must open for pilus biogenesis is trivial*
*because considering the dimensions of the competence pilus gate opening is a prerequisite to*
*extrude the type IV competence pilus.*

*Overall the work appears well done and paper is clearly and concisely written, but there are*
*still some issues that need to be addressed before the manuscript will be acceptable for*
*publication.*

*Page 1, line 27 -28: The structural analyses of the PilQ mutants does not unravell the*
*mechanism of PilQ. That the gate has to open to extrude the type IV competence pilus is*
*evident from the dimensions of the type IV competence pilus and the dimensions of the VcPilQ*
*gate. However the structural analyses do not shed any light onto the mechanism of gate*
*closing and opening, on the signals required, how are these signal are transferred, which other*
*components are involved in signal transfer and how does the secretin interact with its*
*substrates such as the competence pilus or DNA?*

We agree. That sentence has been revised.

*Page 7, line 185: The assignment of the AMIN domain to the hazy density present in the 2D*
*classification in Figure 1B needs to be solidified. The same holds true for the residues 126 – 159*
*following the AMIN domain. Deletion derivatives devoid of the AMIN domain and/or devoid of the*
*region following the AMIN domain in VcPilQ have to be generated and subject to structural*
*analyses.*

We agree that to conclusively identify the AMIN domain in the 2D classes, we would need to
solve the structure of an AMIN deletion derivative. We note however that Koo et al. already
did this experiment in their **2016's analysis of the P. aeruginosa PilQ AMIN deletion** (Figure 2
below). They found that "the two AMIN domains in the full-length protein are either poorly
ordered in solution or denatured by the detergent during purification"(Koo et al. 2016).

Figure 2. Comparison of PilQ_{2xHis8} and PilQ_{Δ26-280-2xHis8} by Western Blot and Negative Stain Electron Microscopy

(A) Western blot for PilQ in *P. aeruginosa pilQ::FRT*, *P. aeruginosa pilQ::FRT* complemented with PilQ_{2xHis8} and *P. aeruginosa pilQ::FRT* complemented with PilQ_{Δ26-280-2xHis8} showing SDS-resistant multimers in the top panel and monomers resulting from phenol dissociation in the bottom panel.

(B) Comparison of PilQ_{2xHis8} and PilQ_{Δ26-280-2xHis8} 2D side view class averages by negative-stain electron microscopy.

cretins do not (Guilvout et al., 2008; Tosi et al., 2014). For those secretins that autoassemble, the secretin domain and the preceding N1 domain are necessary and sufficient. For *P. aeruginosa*, PilQ multi-

If we performed structural analysis on the VcPilQ AMIN domain deletion (and the unstructured
 residues 126-159), we might also see the fuzzy halo disappear from the 2D classes. But if the
 hazy density remained, we would also have to acknowledge that it might be due to damaged
 particles (air/water interface unfolding, proteolysis, His tag, etc). Because this experiment
 would not substantially enhance our understanding of the mechanism of PilQ, we have instead
 simply revised the text to clarify that our assignment of the hazy density to the AMIN domains
 is still just speculation.

*Page 7, line 195: The authors suggest that the AMIN domains are probably not regularly*
 *arranged in situ. The structural data presented do not provide any evidence for this conclusion.*

While our structural data DO clearly show the AMIN domain are not regularly arranged in vitro
 (otherwise they would have been seen), we agree that this is NOT proof that the AMIN
 domains are also unstructured *in situ*. We have therefore revised the text to emphasize that is
 only our hypothesis. It is a very reasonable hypothesis, however, because the AMIN domain is
 thought to interact with the irregular peptidoglycan layer, not laterally with other subunits, so it
 is hard to imagine how they could be regularly ordered in situ. Purified AMIN domains from *E.*
 *coli* AmiC (cell wall enzyme) and the *N. meningitidis* PilQ are also monodisperse, which
 suggests that they don't laterally associate into a ring *in vitro* (Rocaboy et al. 2013; Berry et al.
 2012).

*To get insights into the in situ arrangement of VcPilQ the authors should analyze the in situ*
 *arrangement by cryo tomography of V. cholerae cells producing the His-tagged VcPilQ.*

We agree this will be interesting. Analysis of the *in situ* structure by sub-tomogram averaging
 is an ongoing, long-term project in the Jensen and Dalia labs and will not be completed in time
 for a revision to this manuscript.

*Page 8, line 212: The gate of the secretin has the most narrow inner diameter of the PilQ*
 *channel. Information with respect to the charge of the inner surface of the gate should be*
 *stated in the text.*

Done.

*Page 10, line 261: There is no figure 4C.*

Corrected.

*How was the detergent belt around the putative outer membrane region determined?*

We have now included an enhanced description of this process in the Methods (and see
response above to Reviewer #1).

*Page 11, line 280 – 282: Why do the V. cholerae mutants which are blocked in DNA uptake by*
*locking the gate of the PilQ channel only exhibit a reduced natural transformation phenotype*
*but are not completely defect in natural transformation?*

The lack of an absolute phenotype is because disulfide bond formation is likely not 100%
efficient. Thus, instances where disulfide bonds have not formed between PilQ monomers will
allow for some degree of pilus activity (and corresponding natural transformation). The
manuscript has been revised to clarify this point.

*Page 11, line 282 -283: Statistics on piliation is missing. How many wt and mutant cells were*
*piliated and how many pili were detected per cell?*

As recognized by Reviewer #3 below, as it turns out it is quite difficult to get reliable estimates
of these numbers. Pili are transient, of different lengths, and not always visible in light
microscopes because of where they emanate from the cell and labelling efficiency.
Unfortunately the same problems hamper negative stain EM and cryo-ET, and cryo-ET is
further very time consuming and suffers from the missing wedge effect, which obscures pili in
certain orientations with respect to the tilt axis (see more complete explanation below). We do
not believe knowing the exact number of pili would improve our understanding of the
mechanism of the T4aP and PilQ, however: the transformation assay is a quantitative
demonstration that pilus activity is perturbed by the cysteine pair mutants and that activity is
recovered with reducing agent. We included the piliation data to qualitatively back this point up
by showing that pilus biogenesis is what is affected (as expected and obvious based on what
the reviewers point out). We have revised the manuscript to clarify these points.

***Reviewer #3 (Remarks to the Author):***

*The key result from this paper is the determination of the structure of PilQ from Vibrio, which is*
*achieved (for the majority of the protein) at a resolution sufficient to build an atomic model.*
*There are some interesting differences between the T2SS and T3SS counterparts described.*
*Some cysteine mutagenesis is also carried out in the gate region with experimental analysis of*
*the mutants.*

**Major points:**

1. The authors describe the high-resolution structure of the PilQ secretin from *Vibrio*, which
they describe as “a Type IV competence pilus secretin”. This is confusing because it is not
clear if this is different to the PilQ Type IV secretins in numerous other systems that are not
specifically described as “competence secretins”. If there are genuine differences in the type IV
pilus systems (competence or not) this would be interesting to highlight and clarity should be
provided throughout the entire manuscript e.g. L83-88, L210-219, L249, L378-388, and all
figures where comparisons are made. However, if there are no significant differences, the
authors should clearly state this from the outset in order to clarify the nomenclature. In
attempting to navigate this issue, this author came across a recent Review (Piepenbrink, K. H;
2019; <https://doi.org/10.3389/fmolb.2019.00001>) that describes *Vibrio* as expressing type IV
pili (and being naturally competent like many other
gram negative bacteria with type IV pili), whereas Gram positive bacteria express competence
pili, which are something different altogether - they do not have PilQ.

We agree that the nomenclature can be confusing! To address this point, we performed a
phylogenetic analysis of Type IVa Pilus (T4aP) machine secretins in Proteobacteria and
examined the protein domain architecture (**Figure 4A-B, Supplemental Figure 8**). We
mapped the T4aP systems known to participate in natural transformation onto the tree (**Figure**
**4A**) and concluded that at this point we cannot justify the separation of competence-related
secretins from T4aP secretins. This is in part because we don't have enough functional
information for different T4aP secretins. We don't know conclusively if a given sequence
facilitates natural transformation or not, so we cannot annotate our tree.

Historically, T4P systems have been categorized into a- and b-types based on the sequence of
their major pilin subunit (the protein that makes up the filament we call the pilus)(Craig, Forest,
and Maier 2019; Pelicic 2008). Over the years, the pilus system that we studied here has been
called the Type IVa pilus (T4aP), the Chitin-regulated Pilus (ChiRP), and the Type IV
competence pilus (T4CP)(Meibom et al. 2004; Ellison et al. 2018). Recently, the term Type IVc
pilus has been suggested for tad-type T4P(Ellison, Kan, et al. 2019), so the T4CP abbreviation
for the competence pilus is not ideal.

We initially favored the name “Type IV competence pilus secretin” because *V. cholerae* has
two different sets of T4aP with drastically different functions: The T4aP used for competence
discussed in this paper and the mannose-sensitive hemagglutinin A (MSHA) pilus. In *V.*
*cholerae*, the T4aP for competence is used to take up DNA and may be involved in adherence
and kin recognition(Seitz and Blokesch 2013; Adams et al. 2019), while the MSHA pilus is
used for adherence(Jonson, Holmgren, and Svennerholm 1991). Thus, we wanted to clarify
which *V. cholerae* T4aP secretin we were discussing. The MSHA pilus has a distinct secretin
(MshL).

Both gram negative and gram positive bacteria can have T4P. Of course, since gram positive
bacteria lack an outer membrane, they do not have an outer membrane secretin. The
Piepenbrink review denotes the gram positive T4P as a “competence pilus”, but it is far from
the only T4P involved in competence(Piepenbrink 2019). A variety of gram negative bacteria
use T4P for competence. In *Neisseria gonorrhoeae*, the gonococcal (GC) pilus (a T4aP)
mediates adhesion, twitching motility, and competence, whereas in *Pseudomonas aeruginosa*,
the PAK pilus (a T4aP) is used for adhesion and motility, but not competence (Craig, Pique,
and Tainer 2004; Plant and Jonsson 2006). In *V. cholerae*, the T4aP used for competence

may play a role in adhesion, but does not promote motility(Seitz and Blokesch 2013; Adams et
al. 2019).

We have therefore settled on calling the structure we solved a Type IVa Pilus (T4aP) secretin
and will drop the word competence, but clarify in the Intro that we are talking about the T4aP in
*V. cholerae* responsible for competence, not the MSHA T4aP.

*2. Related to point 1, on L382, the authors go on to say that the “Vibrio Type IV competence*
*pilus is in fact a Type IVa pilus”. This argument is used to justify docking their new Vibrio*
*“competence secretin” PilQ into their previously determined density map for PilQ from the*
*“Type IVa pilus” from Myxococcus.*

Several structures from sub-tomogram averaging of Type IV Pilus (T4P) systems have been
reported including the *Myxococcus xanthus* T4aP, the *V. cholerae* Toxin Co-regulated Pilus
(TCP, a T4bP), and the *Thermus thermophilus* T4P(Gold et al. 2015; Chang et al. 2016; 2017).
We compared VcPilQ (a Type IVa Pilus (T4aP) secretin) to the *M. xanthus* T4aP sub-
tomogram average because it is more similar to VcT4aP than the other systems analyzed by
sub-tomogram averaging. Hopefully our new clarification of names and relationships in the text
will help readers appreciate this point.

*Regardless of the confusion in nomenclature, this simple analysis takes up ~40 lines of text*
*with the outcome being that PilQ may not penetrate the outer membrane fully. This is not*
*written in a way that conveys a particularly exciting result and should be removed or made*
*considerably more succinct. The implications of this should be spelt out more clearly.*

We have now substantially reworked the text and figures to favor brevity. We condensed the
outer membrane thickness discussion to one panel of **Figure 4** and reduced the discussion of
that point to 17 lines (L286-L303). Nevertheless we do think the outcome is very interesting:
as the Reviewers pointed out, it is obvious that the T4a pilus must be able to cross the outer
membrane. **The secretin is thought to mediate the pilus’s passage through the outer**
**membrane.** So if PilQ does not fully penetrate the outer membrane, how is the pilus getting
out? Is another protein involved? Is there a substantial conformational change in the protein *in*
*vivo* as compared to *in vitro*?

As also described above, the thickness of the micelle in the cryoEM structures of T2SS and
T4P secretins that have been solved is always much thinner than it should be compared to real
membranes. This suggests a disconnect between the reality *in situ* and our structural
understanding of secretins.

For *T. thermophilus*, **an additional “crown domain”** was observed in the single particle cryoEM
structure of TtPilQ on the extracellular face of the secretin(**D’Imprima et al. 2017**). Part of
Figure 3 is reproduced below to demonstrate the location of the crown domain. The crown
domain could not be identified by mass spectrometry analysis.

In the image below, we have reproduced two more figures from D'Imprima et al. 2017 to
 demonstrate that the DDM micelle is about half as thick as the real membrane examined in the
 sub-tomogram average.

D'Imprima et al. 2017

The implication is that it is not clear if, or how, PiIQ penetrates the outer membrane. We have
 now tried to spell that point out as clearly as possible.

3. In their model building, the authors use I-TASSER to build a homology model for the less
 well resolved N0 domain which they then dock into their map. The model is found to be similar
 to previous structures, which gives confidence that it is correct – but it is somewhat at odds
 with the considerable words (L299-311) and an entire Figure (Fig. S7) that the authors

*dedicate to describing how homology modelling is generally insufficient to predict structure.*
*Therefore it seems that Fig. S7 and the accompanying text does not add anything to support*
*the manuscript.*

Discussions of homology modeling in the manuscript have been revised and Figure S7 has
been removed.

*4. The authors show that the introduction of disulphide bonds in the gate region impairs natural*
*transformation and piliation. Being as this would mean that the gate cannot move or open, this*
*is not really surprising. Likewise, the section title “Cysteine mutants indicate gate must open for*
*pilus biogenesis and natural transformation” is somewhat obvious. It’s hard to see how a drug*
*would be able to induce such a change (L32). The manuscript would be stronger if the authors*
*could provide more compelling evidence for how they could specifically inhibit gate movement.*

We fully agree it would be great to identify a way to specifically inhibit gate movement – it
could be a great drug lead compound(!) – but this is beyond the scope of this paper. When we
suggest that VcPilQ could be druggable, we imagine that a small molecule could be designed
to bind in the gate region of VcPilQ, not that a drug would generate a disulfide bond. We have
clarified this point in the text, and added that our results confirm others already in the literature.

*5. Why were the cysteine pair mutants made in a PilT deletion stain and not wild-type? As PilT*
*is the retraction ATPase, wouldn’t this mean that there would be an over-inflated number of pili*
*assembled on the cell surface? Is this taken into account?*

We assessed piliation in the Δ pilT background to sensitize the assay to be able to test the
effects of gate locked vs gate open on pilus biogenesis. The dynamic activity of Vc
competence pili is much higher than that described for many other pilus systems. Such that
within a snapshot, very few cells will have surface exposed pili. Thus, to see how gate locking
affects pilus biogenesis, the Δ pilT background is actually a better background to use than the
parent. But for functionality of the pili, we employed a transformation assay where pilT is intact
- this latter assay does not require us to directly look or test the activity of cells within a
snapshot. Thus, the transformation assay can integrate the activity of the highly dynamic pili
over a longer timeframe to test their function.

We have revised the text to clarify these issues and added **Supplemental Table 1** describing
the strains used, including the following:

For the purification of VcPilQ, a 10xHis tag was added at between the signal peptide and PilQ
in the native locus of genome to generate a fully functional, His-tagged PilQ. We
demonstrated it is function in Supplemental **Figure 1A** using a transformation assay.

For **Figure 3D**, the cysteine pair mutants were generated in a similar background as the fully-
functional His-tagged PilQ used for purification, except that for the cysteine pair mutants PilQ
is knocked out at its native locus and expressed ectopically. The control strain (TND2140)
expresses pBAD-10xHis-PilQ, while the cysteine mutants express pBAD-10xHis-PilQ(S448C
S453C) (TND2169) or express pBAD-10xHis-PilQ(L445C T493C) (TND2170). This allowed us
to test the natural transformation abilities of the cysteine pair mutants.

For **Figure 3E-G**, the cysteine pair mutants were generated in a strain with PilA(S67C) and
 Δ PilT. The S67C mutation in the major pilin subunit PilA facilitates fluorescent labeling of the
 pilus. The Dalia lab previously demonstrated that the transformation frequency of fluorescently-
 labeled PilA(S67C) T4a pili is identical to that of unlabeled, wild-type pili (Ellison et al. 2018).
 PilT is the retraction ATPase for the T4aP competence system discussed here, so Δ pilT
 mutants exhibit increased surface piliation (hyperpiliation). The hyperpiliated PilT deletion
 strain is commonly used in the *V. cholerae* natural transformation field to prolong the time
 T4aP are present on the surface of cells (Meibom et al. 2005; Seitz and Blokesch 2013).

*6. The authors assess the level of piliation in their cysteine pair mutants. Fluorescent labelling*
 *and light microscopy imaging doesn't seem particularly accurate for determining the amount of*
 *pili and it's difficult to assess the result. Readers also need to refer to other papers to follow the*
 *methodology. How many cells were tested and how many pili? A higher resolution imaging*
 *technique such as electron microscopy should be used where the pili can be observed directly.*

We agree that diffraction-limited fluorescence microscopy is not a great way to assess the
 number of pili on the surface of a cell. However, in Figure 3 our goal was to show that the
 cysteine mutants exhibit reduced transformation efficiency in the absence of reducing agent
 (**Figure 3D**), and that this reduced transformation efficiency corresponds to few surface pili in
 the absence of reducing agent (**Figure 3E**). The reappearance of pili in the 1 and 2 mM DTT
 conditions (**Figure 3F-G**) demonstrate that the increase in transformation efficiency (**Figure**
 **3D**) is associated with the presence of pili. There is some discussion in the review article that
 Reviewer 3 pointed out earlier that transformation (Figure 7 of Piepenbrink 2019, see below)
 could occur without a pilus (Piepenbrink 2019). Thus, we wanted to know if the increased
 transformation efficiency of the 1 and 2 mM DTT conditions in Figure 3D also showed pili.

Accurately estimating the number of pili per cell by electron microscopy is complicated for
 several reasons. 1) In our experience in the Jensen lab, pili can break off of cells during the
 blotting that occurs just before vitrification, so an estimate of number of pili per cell by

cryogenic electron tomography (cryoET) would be inherently low. 2) Pili are similarly damaged
in negative stain electron microscopy by the drying and blotting process, and the estimate is
further limited by the fidelity of the stain representing individual pili. 3) Additionally, pili need to
be counted in 3D tomograms, not 2D projection images. It is difficult to see pili in individual 2D
projection images, and it is dependent on the angle you view the cell from. For example, a
pilus could be hidden between the surface of the cryoEM grid and the bottom face of the cell.
4) The throughput of cryoET experiments is low (typically 2 to 3 tomograms per hour using a
conventional scheme), the microscope time is expensive, and the analysis is laborious. 5) In
tomography, the resolution comes at the expense of the viewing area. An entire *V. cholerae*
cell won't fit in the viewing area of a tomogram at the resolution needed to accurately
distinguish and count T4P. If the pili were limited to cell poles (like the *P. aeruginosa*
T4aP (Carter et al. 2017)), it would be easier to confidently count them by cryoET (because the
cell pole would fit in a single tomogram). 6) Because of the missing wedge in cryo-ET, pili in
perpendicular to the tilt axis are almost invisible. However, the competence pili in the VcT4aP
discussed in this paper do not display polar localization (Seitz and Blokesch 2013). For these
reasons, obtaining an accurate count of pili per cell using electron microscopy is not trivial.

Therefore, we would like to respectfully assert that the piliation assays remain qualitative for the
scope of this study. The natural transformation assays for the cysteine pair mutants provide a
quantitative measure of transformation efficiency recovery in the presence of reducing agent,
and the qualitative piliation results support that conclusion. We have however revised the
Methods section and the main text to include more details, so readers do not have to refer to
other papers (though they remain referenced).

*7. Related to the above, is there evidence to demonstrate that cysteine mutant PilQ can still*
*assemble in the membranes at wild-type levels?*

Our quantitative natural transformation assays (**Figure 3D**) demonstrate that under reducing
conditions the cysteine pair mutants reach similar levels of natural transformation to the WT
PilQ. Because there is no reason to believe that reducing conditions should specifically alter
the expression / regulation of distinct PilQ alleles in our system, this strongly suggests that
different PilQ cys mutants must have a similar numbers of pilus machines as the parent strain
under the conditions tested.

*8. What is the evidence that the pilus and DNA actually fill the diameter of the secretin channel*
*at the same time? There is a lot of discussion about this point but the evidence for it is not*
*clearly described.*

No one knows how the DNA associates with the *V. cholerae* Type IVa pilus during natural
transformation, so we don't know if they would be present in PilQ simultaneously. No one has
solved the structure of the *V. cholerae* Type IVa pilus used for natural competence.

In 2018, the Dalia lab demonstrated that double stranded DNA mainly binds the tip of the
VcT4a pilus (Ellison et al. 2018). Figure 4E from Ellison *et al.* 2018 has been reproduced
below to show a hypothesized schematic of the retraction of a T4aP with DNA bound at the tip:

Figure caption: “Figure 4.E.: A model of pilus retraction-mediated DNA uptake. Retraction of DNA-
 bound pili threads dsDNA across the outer membrane (OM; left) followed by ComEA-dependent
 molecular ratcheting (right) to promote uptake. IM, inner membrane; PG, peptidoglycan.”(Ellison et al.
 2018)

If Ellison’s interpretation is correct, DNA and the pilus would not need to be present in PilQ
 simultaneously.

On the other hand, in other bacteria (*Neisseria gonorrhoeae* and *Thermus thermophilus*)
 cryoEM structures of the Type IV pilus have been solved(Craig et al. 2006; Neuhaus et al.
 2020). Based on the electrostatics of the pilus surface, it has been suggested that DNA may
 wind around the Type IV Pilus(Craig et al. 2006; Neuhaus et al. 2020). For example,
 Supplementary Figure 9 from Neuhaus et al. 2020 is reproduced below to show hypothesized
 binding of DNA to a groove in the pilus(Neuhaus et al. 2020):

Supplementary Figure 9: Model of DNA bound to wide pilus

A double stranded DNA molecule (green) is modelled around a wide pilus shown in surface charge representation (negative charges, red; positive charges, blue). The DNA backbone fits neatly into the positively charged groove of the PilA4 filament (inset). Post-translational modifications are shown in yellow (transparent yellow in inset). Scale bar, 10 Å.

If DNA winds around the pilus, PilQ would need to accommodate both the pilus and DNA
 simultaneously.

How bulky can the pilus become and still pass through PilQ? It is not clear, but several papers
have touched on this idea. For example, in *Neisseria meningitidis*, pilus formation is inhibited
when the pilin protein PILE is fused to mCherry(Imhaus and Duménil 2014). The Dalia lab has
also demonstrated that pilus retraction can be inhibited by maleimide-conjugated molecules
(like maleimide-conjugated PEG5000 and or biotin-maleimide followed by the NeutrAvidin
protein) in the *V. cholerae* T4aP, the *V. cholerae* T4bP (the toxin co-regulated pilus, TCP), and
the *Caulobacter crescentus* Tad pilus(Ellison, Dalia, et al. 2019).

Ongoing work in the Dalia and Jensen labs is now attempting to address this question. The
manuscript has been revised to clarify all these points.

*9. There is a vast amount of text (L378-416; some 38 lines) about how the PilQ structure could*
*accommodate a pilus. Being as this is the role of PilQ, it's not particularly surprising and should*
*be written more succinctly.*

This section has been revised to be more concise (~21 lines)

*10. Many of the figures should be combined to generate a more succinct manuscript. E.g. Fig.*
*2 and 4 do need to stand-alone and Fig. 5 and 6 are for the same experiment. Fig. 7 should be*
*supplementary.*

OK. Several panels have been removed and the remaining figures streamlined to create a
more succinct manuscript with four main figures

***Minor points:***

*1. On L65, the authors describe how chitin on the exoskeletons of crustaceans induces*
*expression of the machinery. How this links to human disease is not made very clear.*

The text has been revised to clarify this point.

*2. An AMIN domain is not explained. As the flexibility of these domains is used to justify a lack*
*of observed density in the cryoEM maps (L194), it should be properly described.*

We have revised the text to clarify this point.

We would like to clarify that we think the VcPilQ AMIN domain itself (residues 54-125) likely
adopts a fold similar to the purified AMIN domains from *E. coli* AmiC (cell wall enzyme) and the
*N. meningitidis* PilQ (Rocaboy et al. 2013; Berry et al. 2012). Because we did not see any
secondary structure for the AMIN domain in our single particle structure, we hypothesize that
the link between the AMIN domain and the N0 domain (linker residues 126-159, N0 domain
starting at residue 160) is probably flexible.

Here is a mockup of what we expect the structure looks like with the Rocaboy 2013 AmiC AMIN
domain (PDB 4BIN, red or grey below) included:

3. The authors mention and show in Fig. 2 work from Koo et al and D'Imprima et al, but these could not be found in the bibliography.

Thank you for pointing out this mistake. It has been corrected and we have also checked all references to ensure they are correct.

4. The description of cryoEM sample preparation and imaging conditions adds an unnecessary 20 lines to the Results section (L159-179). It's already described in Methods and Fig. S4.

We have modified the manuscript to be more concise.

5. Is the reference to Fig. S8 on L206 correct?

Corrected.

6. Why are there so many gels in Fig. S1? They appear to show the same thing.

This figure has been simplified to show a single representative experiment.

Bibliography:

- Adams, David. W., Sandrine Stutzmann, Candice Stoudmann, and Melanie Blokesch. 2019. “DNA-
Uptake Pili of *Vibrio Cholerae* Are Required for Chitin Colonization and Capable of Kin
Recognition via Sequence-Specific Self-Interaction.” *Nature Microbiology* 4 (9): 1545–57.
<https://doi.org/10.1038/s41564-019-0479-5>.
- Berry, Jamie-Lee, Marie M. Phelan, Richard F. Collins, Tomas Adomavicius, Tone Tønjum, Stefan A.
Frye, Louise Bird, et al. 2012. “Structure and Assembly of a Trans-Periplasmic Channel for Type
IV Pili in *Neisseria Meningitidis*.” *PLOS Pathogens* 8 (9): e1002923.
<https://doi.org/10.1371/journal.ppat.1002923>.
- Carter, Tyson, Ryan N. C. Buensuceso, Stephanie Tammam, Ryan P. Lamers, Hanjeong Harvey, P.
Lynne Howell, and Lori L. Burrows. 2017. “The Type IVa Pilus Machinery Is Recruited to Sites
of Future Cell Division.” *MBio* 8 (1): e02103–16. <https://doi.org/10.1128/mbio.02103-16>.
- Chang, Yi-Wei, Andreas Kjær, Davi R Ortega, Gabriela Kovacicova, John A Sutherland, Lee A
Rettberg, Ronald K Taylor, and Grant J Jensen. 2017. “Architecture of the *Vibrio Cholerae*
Toxin-Coregulated Pilus Machine Revealed by Electron Cryotomography.” *Nature Microbiology*
2 (4): 16269. <https://doi.org/10.1038/nmicrobiol.2016.269>.
- Chang, Yi-Wei, Lee A Rettberg, Anke Treuner-Lange, Janet Iwasa, Lotte Søgaard-Andersen, and Grant
J Jensen. 2016. “Architecture of the Type IVa Pilus Machine.” *Science* 351: aad2001.
<https://doi.org/10.1126/science.aad2001>.
- Craig, Lisa, Katrina T Forest, and Berenike Maier. 2019. “Type IV Pili: Dynamics, Biophysics and
Functional Consequences.” *Nature Reviews Microbiology*, 1–12. <https://doi.org/10.1038/s41579-019-0195-4>.
- Craig, Lisa, Michael E Pique, and John A Tainer. 2004. “Type IV Pilus Structure and Bacterial
Pathogenicity.” *Nature Reviews Microbiology* 2 (5): 363. <https://doi.org/10.1038/nrmicro885>.
- Craig, Lisa, Niels Volkmann, Andrew S Arvai, Michael E Pique, Mark Yeager, Edward H Egelman, and
John A Tainer. 2006. “Type IV Pilus Structure by Cryo-Electron Microscopy and
Crystallography: Implications for Pilus Assembly and Functions.” *Molecular Cell* 23 (5): 651–
662. <https://doi.org/10.1016/j.molcel.2006.07.004>.
- D’Imprima, Edoardo, Ralf Salzer, Ramachandra M Bhaskara, Ricardo Sánchez, Ilona Rose, Lennart
Kirchner, Gerhard Hummer, Werner Kühlbrandt, Janet Vonck, and Beate Averhoff. 2017.
“Cryo-EM Structure of the Bifunctional Secretin Complex of *Thermus Thermophilus*.” *ELife* 6:
e30483. <https://doi.org/10.7554/elife.30483>.
- Ellison, Courtney K, Triana N Dalia, Alfredo Ceballos, Joseph Wang, Nicolas Biais, Yves V Brun, and
Ankur B Dalia. 2018. “Retraction of DNA-Bound Type IV Competence Pili Initiates DNA
Uptake during Natural Transformation in *Vibrio Cholerae*.” *Nature Microbiology* 3 (7): 773–
780. <https://doi.org/10.1038/s41564-018-0174-y>.
- Ellison, Courtney K, Triana N Dalia, Ankur B Dalia, and Yves V Brun. 2019. “Real-Time Microscopy
and Physical Perturbation of Bacterial Pili Using Maleimide-Conjugated Molecules.” *Nature*
*Protocols* 14 (6): 1–17. <https://doi.org/10.1038/s41596-019-0162-6>.
- Ellison, Courtney K., Jingbo Kan, Jennifer L. Chlebek, Katherine R. Hummels, Gaël Panis, Patrick H.
Viollier, Nicolas Biais, Ankur B. Dalia, and Yves V. Brun. 2019. “A Bifunctional ATPase
Drives Tad Pilus Extension and Retraction.” *Science Advances* 5 (12): eaay2591.
<https://doi.org/10.1126/sciadv.aay2591>.
- Ghosal, Debnath, Ki Woo Kim, Huaixin Zheng, Mohammed Kaplan, Hilary K. Truchan, Alberto E.
Lopez, Ian E. McIntire, Joseph P. Vogel, Nicholas P. Cianciotto, and Grant J. Jensen. 2019. “In
Vivo Structure of the *Legionella* Type II Secretion System by Electron Cryotomography.”
*Nature Microbiology* 4 (12): 2101–8. <https://doi.org/10.1038/s41564-019-0603-6>.
- Gold, Vicki AM, Ralf Salzer, Beate Averhoff, and Werner Kühlbrandt. 2015. “Structure of a Type IV
Pilus Machinery in the Open and Closed State.” *ELife* 4: e07380.
<https://doi.org/10.7554/elife.07380>.

- Hay, Iain D, Matthew J Belousoff, Rhys A Dunstan, Rebecca S Bamert, and Trevor Lithgow. 2018.
“Structure and Membrane Topography of the Vibrio-Type Secretin Complex from the Type 2
Secretion System of Enteropathogenic Escherichia Coli.” *Journal of Bacteriology* 200.
<https://doi.org/10.1128/JB.00521-17>.
- Hay, Iain D., Matthew J. Belousoff, and Trevor Lithgow. 2017. “Structural Basis of Type 2 Secretion
System Engagement between the Inner and Outer Bacterial Membranes.” *MBio* 8 (5): e01344–
17. <https://doi.org/10.1128/mbio.01344-17>.
- Howard, Peter S, Leandro F Estrozi, Quentin Bertrand, Carlos Contreras-Martel, Timothy Strozen,
Viviana Job, Alexandre Martins, Daphna Fenel, Guy Schoehn, and Andréa Dessen. 2019.
“Structure and Assembly of Pilotin-Dependent and -Independent Secretins of the Type II
Secretion System.” *PLOS Pathogens* 15 (5): e1007731.
<https://doi.org/10.1371/journal.ppat.1007731>.
- Imhaus, Anne-Flore, and Guillaume Duménil. 2014. “The Number of Neisseria Meningitidis Type IV
Pili Determines Host Cell Interaction.” *The EMBO Journal* 33 (16): 1767–83.
<https://doi.org/10.15252/embj.201488031>.
- Jonson, G., J. Holmgren, and A. M. Svennerholm. 1991. “Identification of a Mannose-Binding Pilus on
Vibrio Cholerae El Tor.” *Microbial Pathogenesis* 11 (6): 433–41. [https://doi.org/10.1016/0882-4010\(91\)90039-d](https://doi.org/10.1016/0882-4010(91)90039-d).
- Koo, Jason, Ryan P Lamers, John L Rubinstein, Lori L Burrows, and Lynne P Howell. 2016. “Structure
of the Pseudomonas Aeruginosa Type IVa Pilus Secretin at 7.4 Å.” *Structure* 24 (10): 1778–87.
<https://doi.org/10.1016/j.str.2016.08.007>.
- Meibom, Karin L, Melanie Blokesch, Nadia A Dolganov, Cheng-Yen Wu, and Gary K Schoolnik. 2005.
“Chitin Induces Natural Competence in *Vibrio Cholerae*.” *Science* 310 (5755): 1824–1827.
<https://doi.org/10.1126/science.1120096>.
- Meibom, Karin L, Xibing B Li, Alex T Nielsen, Cheng-Yen Wu, Saul Roseman, and Gary K Schoolnik.
2004. “The Vibrio Cholerae Chitin Utilization Program.” *Proceedings of the National Academy
of Sciences of the United States of America* 101 (8): 2524–2529.
<https://doi.org/10.1073/pnas.0308707101>.
- Metzger, Lisa C, and Melanie Blokesch. 2014. “Composition of the DNA-Uptake Complex of Vibrio
Cholerae.” *Mobile Genetic Elements* 4 (1): e28142. <https://doi.org/10.4161/mge.28142>.
- Neuhaus, Alexander, Muniyandi Selvaraj, Ralf Salzer, Julian D. Langer, Kerstin Kruse, Lennart
Kirchner, Kelly Sanders, Bertram Daum, Beate Averhoff, and Vicki A. M. Gold. 2020. “Cryo-
Electron Microscopy Reveals Two Distinct Type IV Pili Assembled by the Same Bacterium.”
*Nature Communications* 11 (1): 2231. <https://doi.org/10.1038/s41467-020-15650-w>.
- Pelicic, Vladimir. 2008. “Type IV Pili: E Pluribus Unum?” *Molecular Microbiology* 68 (4): 827–837.
<https://doi.org/10.1111/j.1365-2958.2008.06197.x>.
- Piepenbrink, Kurt H. 2019. “DNA Uptake by Type IV Filaments.” *Frontiers in Molecular Biosciences*
6: 1. <https://doi.org/10.3389/fmolb.2019.00001>.
- Plant, Laura J., and Ann-Beth Jonsson. 2006. “Type IV Pili of Neisseria Gonorrhoeae Influence the
Activation of Human CD4+ T Cells.” *Infection and Immunity* 74 (1): 442–48.
<https://doi.org/10.1128/IAI.74.1.442-448.2006>.
- Rocaboy, Mathieu, Raphael Herman, Eric Sauvage, Han Remaut, Kristof Moonens, Mohammed Terrak,
Paulette Charlier, and Frederic Kerff. 2013. “The Crystal Structure of the Cell Division Amidase
AmiC Reveals the Fold of the AMIN Domain, a New Peptidoglycan Binding Domain: Crystal
Structure of AmiC of *Escherichia Coli*.” *Molecular Microbiology*, September, n/a-n/a.
<https://doi.org/10.1111/mmi.12361>.
- Seitz, Patrick, and Melanie Blokesch. 2013. “DNA-Uptake Machinery of Naturally Competent Vibrio
Cholerae.” *Proceedings of the National Academy of Sciences* 110 (44): 17987–92.
<https://doi.org/10.1073/pnas.1315647110>.

Yan, Zhaofeng, Meng Yin, Dandan Xu, Yongqun Zhu, and Xueming Li. 2017. “Structural Insights into
the Secretin Translocation Channel in the Type II Secretion System.” *Nature Structural &*
*Molecular Biology* 24 (2): 177–83. <https://doi.org/10.1038/nsmb.3350>.

REVIEWERS' COMMENTS

Reviewer #1 (Remarks to the Author):

The authors have made responses to all comments. Corresponding revisions were made in the new manuscript. My concerns are addressed. The manuscript is in a good shape and quality to be published.

Reviewer #2 (Remarks to the Author):

The authors have paid regard to all major points of the reviewer . There are no additional major shortcomings.

Reviewer #3 (Remarks to the Author):

Response to revised manuscript according to original points made

Points 1 and 2: Answered coherently and in great detail and as such the manuscript is much improved. In particular the importance of membrane penetration is clearer. This could be even stronger by including a short discussion around the "crown domain" that is so well described in the rebuttal but is only hinted at in the actual manuscript. Could there be something similar in *V. cholerae* for example?

Points 3, 4 & 5 – ok

Point 6 – The arguments that EM has not been used for the purposes of this study are acceptable when considering that the assessment is qualitative rather than quantitative. However, the sample size and number of cells, plus the number of pili that can be clearly discriminated should be provided in the manuscript somewhere (as a supplementary table / in the legend / in the methods) and reporting summary.

Points 7-10 – ok

Minor points

Point 1 - ok

Point 2 – I am still missing a definition of an AMIN domain (Amidase N-terminal domain?).

Points 3-6 - ok

Final revisions for Nature Communications manuscript NCOMMS-20-09348A

REVIEWERS' COMMENTS

Reviewer #1 (Remarks to the Author):

The authors have made responses to all comments. Corresponding revisions were made in the new manuscript. My concerns are addressed. The manuscript is in a good shape and quality to be published.

Reviewer #2 (Remarks to the Author):

The authors have paid regard to all major points of the reviewer . There are no additional major shortcomings.

Reviewer #3 (Remarks to the Author):

Response to revised manuscript according to original points made

Points 1 and 2: Answered coherently and in great detail and as such the manuscript is much improved. In particular the importance of membrane penetration is clearer. This could be even stronger by including a short discussion around the “crown domain” that is so well described in the rebuttal but is only hinted at in the actual manuscript. Could there be something similar in *V. cholerae* for example?

We have revised the manuscript discussion to mention the crown domain in *Thermus thermophilus* PilQ.

Points 3, 4 & 5 – ok

Point 6 – The arguments that EM has not been used for the purposes of this study are acceptable when considering that the assessment is qualitative rather than quantitative. However, the sample size and number of cells, plus the number of pili that can be clearly discriminated should be provided in the manuscript somewhere (as a supplementary table / in the legend / in the methods) and reporting summary.

We have revised the manuscript to state that more than 200 cells were imaged per condition in the methods and figure legend for the competence pilus labeling experiment.

Points 7-10 – ok

Minor points

Point 1 - ok

Point 2 – I am still missing a definition of an AMIN domain (Amidase N-terminal domain?).

We have revised the manuscript to describe the AMIN domain.

Points 3-6 - ok